# CYK4 relaxes the bias in the off-axis motion by MKLP1 kinesin-6

Yohei Maruyama[1], Mitsuhiro Sugawa[1,2], Shin Yamaguchi[1], Tim Davies [3,6], Toshihisa Osaki [4], Takuya Kobayashi[1], Masahiko Yamagishi [1,2], Shoji Takeuchi [4,5], Masanori Mishima [3✉] & Junichiro Yajima [1,2,5✉]

Centralspindlin, a complex of the MKLP1 kinesin-6 and CYK4 GAP subunits, plays key roles in metazoan cytokinesis. CYK4-binding to the long neck region of MKLP1 restricts the configuration of the two MKLP1 motor domains in the centralspindlin. However, it is unclear how the CYK4-binding modulates the interaction of MKLP1 with a microtubule. Here, we performed three-dimensional nanometry of a microbead coated with multiple MKLP1 molecules on a freely suspended microtubule. We found that beads driven by dimeric MKLP1 exhibited persistently left-handed helical trajectories around the microtubule axis, indicating torque generation. By contrast, centralspindlin, like monomeric MKLP1, showed similarly left-handed but less persistent helical movement with occasional rightward movements. Analysis of the fluctuating helical movement indicated that the MKLP1 stochastically makes off-axis motions biased towards the protofilament on the left. CYK4-binding to the neck domains in MKLP1 enables more flexible off-axis motion of centralspindlin, which would help to avoid obstacles along crowded spindle microtubules.

[1] Department of Life Sciences, Graduate School of Arts and Sciences, The University of Tokyo, Meguro-ku, Tokyo, Japan. [2] Komaba Institute for Science, The University of Tokyo, Meguro-ku, Tokyo, Japan. [3] Centre for Mechanochemical Cell Biology and Division of Biomedical Sciences, Warwick Medical School, University of Warwick, Coventry, UK. [4] Institute of Industrial Science, The University of Tokyo, Meguro-ku, Tokyo, Japan. [5] Research Center for complex Systems Biology, The University of Tokyo, Meguro-ku, Tokyo, Japan. [6] Present address: Department of Biosciences, Durham University, Durham, UK. ✉email: M.Mishima@warwick.ac.uk; yajima@bio.c.u-tokyo.ac.jp

In many kinesins with an N-terminal motor domain (N-kinesins), the ~15 amino acid neck-linker, which links the microtubule-binding catalytic core to the neck coiled coil, has been shown to play crucial roles in the mechanochemical transduction of the chemical energy released by ATP hydrolysis to the mechanical works such as alternate stepping on a microtubule[1,2]. However, this design is not conserved in all the N-kinesins. While N-kinesin such as kinesins-1, -2, -3, -4, -5, -7, -8, -9, and -12 have the conserved sequence feature of the neck-linker, the rest of the N-kinesins including kinesins-6, -10 and -11 lack the conserved sequence and instead have a long neck sequence with many helix-breaking proline residues between the catalytic core and the predicted coiled coil[3]. Note that this region might not be entirely elongated but folded into a compact structure (see below). Kinesin-6 is a prototype of these atypical N-kinesins with a long neck (≥ ~70 amino acids)[4–6].

Centralspindlin is a heterotetrameric protein consisting of a dimer of the kinesin-6 motor subunit MKLP1 (collectively referring to the orthologs of mammalian MKLP1/KIF23, such as *Caenorhabditis elegans* ZEN-4 and *Drosophila* Pavarotti), and a dimer of the non-motor subunit, CYK4 (collectively referring to the orthologs of mammalian RACGAP1/HsMgcRacGAP/HsCYK-4, such as *C. elegans* CYK-4, and *Drosophila* Rac-GAP50C/Tumbleweed)[4,7]. Centralspindlin plays crucial roles in animal cytokinesis both as a key organizer of the post-anaphase microtubule structures and as an important hub for cytokinesis signaling[8–10]. A mutation of human KIF23 is responsible for congenital dyserythropoietic anemia type III[11]. For the functions of centralspindlin in cytokinesis, its timely and sharp accumulation to the center of the central spindle and the midbody is essential[12–15]. This relies on its plus-end directed motility, which is modulated by the CYK-binding to the neck of MKLP1 and by further assembly of the heterotetramers into multimeric clusters. A part of the MKLP1 neck is folded into a globular domain, keeping the inter-head distance within the dimeric form moderate (~13 nm)[3]. CYK4-binding to the neck reconfigures the motor heads in the heterotetrameric centralspindlin suitable for anti-parallel bundling[3]. It also slows down the motility of the clustered MKLP1[3] and the microtubule gliding in the surface gliding assay[3,16]. Higher-order clustering promotes the plus-end directed transport without detachment from the microtubule and is essential for timely equatorial accumulation, which is crucial for cytokinesis signaling[14].

The mitotic spindle is a highly crowded environment for microtubule motors since there are a lot of obstacles such as microtubule-associated proteins (MAPs) including microtubule cross-linkers, and other motors carrying various cargos[17,18]. This is a potential difficulty for the efficient equatorial accumulation of centralspindlin. One possibility to bypass these obstacles on the microtubule track would be to flexibly switch the protofilament. In in vitro assays, occasional protofilament switching results in a 3D motion around the microtubule, in particular, a helical motion when there is a bias in the direction of the off-axis motion. Such 3D motions around the microtubule have been observed for some kinesins with the canonical neck linker (kinesins-2 and -8)[19,20].

Here we investigate whether a team of MKLP1 kinesin-6 motors can switch protofilaments during its plus-end-directed motion by 3D nanometry of the motion of a bead coated with multiple motors on a freely suspended microtubule. We report that dimeric *C. elegans* MKLP1 drives a left-handed helical motion around the microtubule with minimum protofilament switching to the right side while the left-handed helical motions driven by monomers or by the heterotetramers with CYK4 are less persistent, showing more frequent off-axis motion to the right side. We discuss the implications of these observations for the efficient motility in the crowded mitotic spindle.

## Results

### Both the MKLP1 dimer and the centralspindlin heterotetramer display left-handed helical motions around a freely suspended microtubule.

To gain insight into the protofilament switching during the plus-end-directed motion of centralspindlin, we examined the 3D motion of a bead driven by multiple MKLP1 kinesin-6. To examine the influence of CYK4-binding to the atypically long neck region of MKLP1, we compared three MKLP1 constructs: a dimeric MKLP1 kinesin-6 construct ($M_2$, 1–555 a.a. of *C. elegans* ZEN-4), a heterotetrameric centralspindlin complex ($M_2C_2$), comprising $M_2$ and a dimeric CYK4 N-terminal fragment ($C_2$, 1–120 a.a. of *C. elegans* CYK-4), and a monomeric MKLP1 kinesin-6 construct ($M_1$, 1–439 a.a. of *C. elegans* ZEN-4) (Fig. 1a, b). They have a His$_6$-tag for purification and a SNAP-tag for immobilization on a bead at their C-terminus. Our experimental setup with a microtubule suspended away from the coverglass surface allows a bead (0.22 μm diameter) coated with motors to freely move all around the entire surface of the microtubule lattice (Fig. 1c). The 3D motion of the bead was monitored using a three dimensional prismatic optical tracking (termed *tPOT*) microscope[21,22] (Fig. 1d, e and Supplementary Movie 1). Our system with a temperature control unit stably achieves nanometer-scale accuracies in 3D positioning of a bead (the precision for the longitudinal axis along the microtubule was 3 nm, and that for the rotation around the microtubule was 5–7.5 degrees, 0.1 s intervals, Supplementary Fig. 1).

In a preliminary screening for a condition that allows us to observe stable motion of the bead along the microtubule, we realized that the bead needed to be coated with multiple motors. This is consistent with the short run lengths of the MKLP1 constructs when they are not assembled into the higher-order clusters[3,14]. We hereafter used the minimum motor:bead ratio that allowed a stable motion although we don't know exactly how many motors are simultaneously interacting with a microtubule at any given time. Considering the high affinity between MKLP1 and CYK4 ($K_D$ ~ 7 nM)[23], it is likely that the majority of MKLP1 on the $M_2C_2$-coated beads are kept associated with CYK4 in our experimental condition. We observed that both the $M_2$-coated and $M_2C_2$-coated beads moved in left-handed helical trajectories around the microtubule though with different helical pitches ($M_2$: 0.8 ± 0.1 μm, $M_2C_2$: 1.5 ± 0.8 μm, (mean ± standard deviation, SD) (Fig. 2a and Supplementary Fig. 2). The handedness of the helical trajectories by MKLP1 was the same as that by other plus-end directed kinesins (when off-axis motion was detected) such as monomeric kinesin-1[24,25], kinesin-2[19,26], kinesin-3[27], kinesin-5[21], and kinesin-8[20,28], and the opposite to that by a minus end-directed kinesin (kinesin-14[29,30]).

Our preparation of microtubules is a 5:3 mixture of microtubules made of 13-protofilaments and those of 14-protofilaments[21]. While protofilaments in a 13-protofilament microtubule are parallel to the axis of the microtubule, in a 14-protofilament microtubule, they are twisted in a left-handed helix around the microtubule axis with a long helical pitch (6–8 μm)[31]. If an MKLP1-coated bead accurately tracked a single protofilament without protofilament switching like double-headed kinesin-1[31–34], it would have shown a helical trajectory with a pitch and handedness identical to the supertwist of the protofilaments around the microtubule axis, i.e., no helicity on a 13-protofilament microtubule or a left-handed helix with a much longer pitch on a 14-protofilament one. On the other hand, if every step was made onto the binding site at the left-front on the microtubule lattice, the helical pitch would be ~0.1 μm (~ 13 × 8 nm). The observed helical pitches of the MKLP1-driven bead motions were between these values (Fig. 2b and Table 1). This indicates that neither the supertwist of the protofilaments nor invariable left-front stepping alone can explain the observed helical motion driven by MKLP1. Rather it suggests that MKLP1 kinesin-6

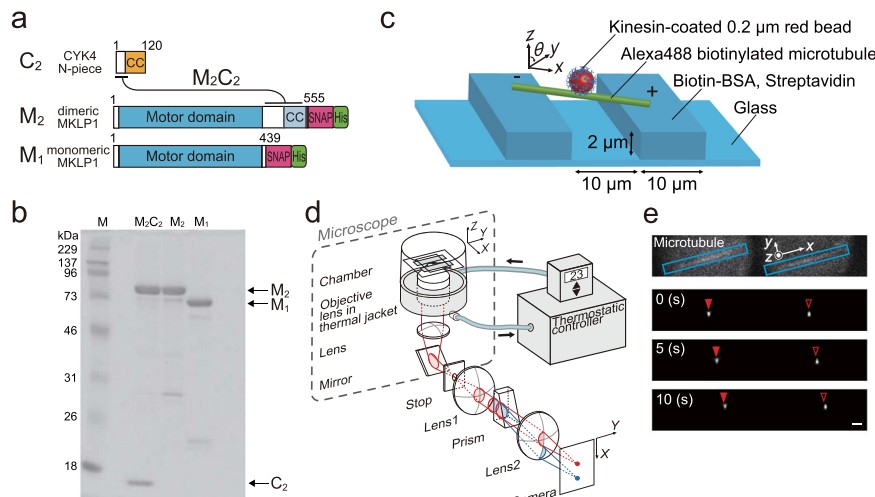

**Fig. 1 3D tracking of MKLP1 kinesin-6 movement around the lattice surface of freely suspended microtubules. a** A schematic of the domain structures of the dimeric N-terminal piece of CYK4 ($C_2$, 1–120 a.a. of *C. elegans* CYK-4), dimeric MKLP1 kinesin-6 ($M_2$, ZEN-4 (1–555 a.a.)-SNAP tag-His tag), monomeric MKLP1 ($M_1$, ZEN-4 (1–439 a.a.)-SNAP tag-His tag), and $M_2C_2$ (the complex of $M_2$ and $C_2$). CC: coiled coil. **b** SDS-PAGE analysis of the purified $M_2$, $M_2C_2$ and $M_1$. Coomassie blue-stained SDS-polyacrylamide gel (12%). Lane 1: molecular size markers (in kDa), Lanes 2–4: purified $M_2C_2$, $M_2$, and $M_1$. The arrows indicate the positions of the individual proteins. **c** A schematic of the experimental setup. Glass was patterned into 2 µm-high, 10 µm-wide parallel walls with a 10 µm gap. Microtubules were suspended between them using streptavidin-biotin interaction. This allowed beads coated with multiple kinesin molecules to move all around the microtubule surface. **d** A schematic drawing of a 3D prismatic optical tracking (termed *tPOT*) microscope. The *z* position of kinesin-coated bead motion as well as the *x–y* position are obtained from a pair of images split by the prism. The temperature of the sample is maintained at 23 °C by the combined temperature management unit. **e** Pairs of split images by the *tPOT* microscope. The top panel shows a pair of split images of an Alexa488-biotin-labeled microtubule (marked by blue rectangles). The other panels show the time series of paired images of an $M_2$-coated bead moving along the suspended microtubule (pairs of solid and open red arrowheads). (Scale bar, 1.5 µm; 5 s interval between images).

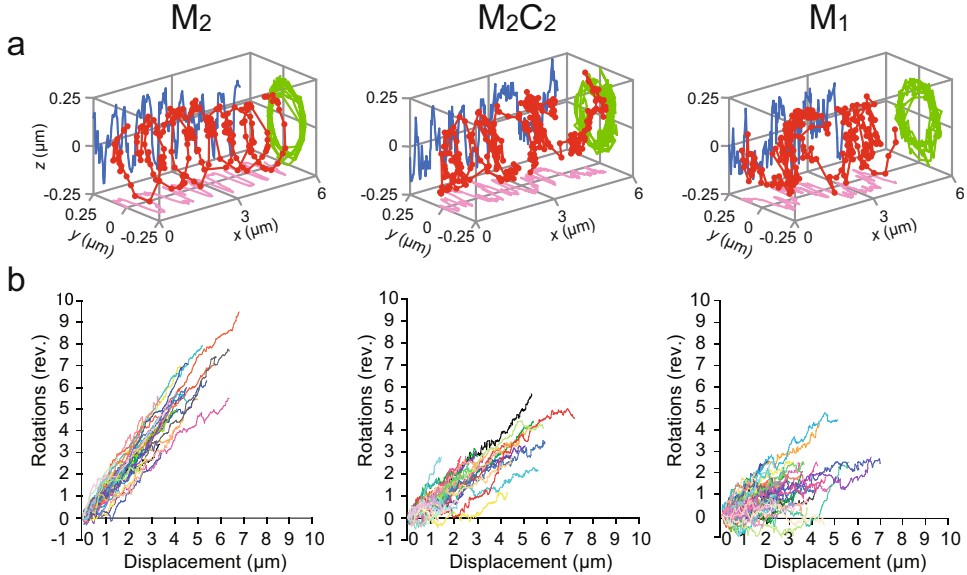

**Fig. 2 3D trajectories of MKLP1 kinesin-6 along the suspended microtubule. a** 3D traces (red) of $M_2$- (left), $M_2C_2$- (middle) and $M_1$- (right) coated beads along a suspended microtubule. The traces exhibited left-handed helical motion along the microtubule. The beads coated with multiple $M_2C_2$ and $M_1$ occasionally reversed their rotational direction for a short period of time. *X–y* (pink), *x–z* (blue), and *y–z* (green) traces are shown. **b** Cumulative revolutions plotted versus the longitudinal displacement for $M_2$ (left), $M_2C_2$ (middle) and $M_1$ (right). Rotation to the left with respect to the movement in the longitudinal direction is expressed with a positive number. Individual traces were fitted with linear functions to obtain the inverse pitch. The average inverse pitch ($M_2$: $1.3 \pm 0.2$ µm$^{-1}$ for 36 beads, $M_2C_2$: $0.7 \pm 0.4$ µm$^{-1}$ for 47 beads, $M_1$: $0.5 \pm 0.4$ µm$^{-1}$ for 47 beads (mean ± SD)) corresponds to the rotational pitch ($M_2$: $0.8 \pm 0.1$ µm, $M_2C_2$: $1.5 \pm 0.8$ µm, $M_1$: $2.1 \pm 1.6$ µm (mean ± SD)). The beads coated with multiple $M_2C_2$ or $M_1$ more frequently exhibited longer pitches than the beads coated with $M_2$ due to occasional reversal of the rotational direction.

**Table 1 Summary of helical motions and microtubule-activated ATPase activity.**

| Construct | Suspended microtubule | | | Non-suspended microtubule | Microtubule activated ATPase | |
|---|---|---|---|---|---|---|
| | Pitch (μm) | Longitudinal velocity (μm s⁻¹) | Rotational velocity (rev s⁻¹) | Longitudinal velocity (μm s⁻¹) | $k_{cat}$ (s⁻¹ head⁻¹) | $K_M$ (μM) |
| $M_2$ | 0.8 ± 0.1 (n = 36) | 0.30 ± 0.03 (n = 36) | 0.37 ± 0.08 (n = 36) | 0.21 ± 0.04 (n = 12) | 39 ± 1 | 0.4 ± 0.04 |
| $M_2C_2$ | 1.5 ± 0.8 (n = 47) | 0.19 ± 0.05 (n = 47) | 0.12 ± 0.05 (n = 47) | 0.15 ± 0.05 (n = 10) | 45 ± 1 | 1.6 ± 0.1 |
| $M_1$ | 2.1 ± 1.6 (n = 47) | 0.24 ± 0.04 (n = 47) | 0.11 ± 0.09 (n = 47) | N.E. | 37 ± 4 | 7.1 ± 1.4 |

In assays using either dimeric $M_2$, centralspindlin complex $M_2C_2$ or monomeric $M_1$ coated-beads, only beads that moved more than ~1 μm were analyzed. The positive values in rotational velocity indicate that the helical motion is left-handed. Data are given as mean ± SD. $n$, number of beads. In the ATPase assays, the microtubule-stimulated ATPase activities were measured from three independent experiments under the same conditions. The turn-over rate ($k_{cat}$) and Michaelis constant for microtubules ($K_M$) were obtained from best fit by the Michaelis–Menten equation.

exerts a mechanical force that consists of not only the axial component along a protofilament, but also the torque-generating, off-axis component, which causes occasional protofilament switching. Note that this torque is distinct from the one that causes the rotation/spin of the marker attached to the tail of dimeric kinesin-1[35].

**CYK4-binding influences the pitch of the helical motion by MKLP1 kinesin-6.** Interestingly, the helical motion of the bead driven by $M_2C_2$ fluctuated more than that driven by $M_2$, although its helicity was still predominantly left-handed (Fig. 2a left and middle). On the plot of rotation vs longitudinal displacement, the trajectories of the beads coated with $M_2$ were found within a relatively narrow zone, showing that the motions in the two directions are well correlated (Fig. 2b left). This is in contrast to the trajectories of the beads coated with $M_2C_2$, which exhibited larger fluctuations (Fig. 2b middle). $M_2$ and $M_2C_2$ also showed a significant difference in the average pitch of the helical motion (0.8 ± 0.1 μm and 1.5 ± 0.8 μm (mean ± SD), $p < 0.001$, Kruskal-Wallis test with correction for multiple comparisons). This difference in the helical pitch suggests that CYK4-binding to the neck of MKLP1 reduces the left-sided bias in the direction of the force generated by the MKLP1 dimer.

Separate plots of the time development of the longitudinal displacement and the revolution (Fig. 3) showed that the larger fluctuation in the $M_2C_2$ trajectories was mainly due to the larger fluctuations of $M_2C_2$ in the rotational movement than those in the longitudinal movement. While, along the longitudinal axis, both $M_2$ and $M_2C_2$ showed largely persistent, unidirectional movements (Fig. 3a left and middle), they (especially, $M_2C_2$) displayed pauses and reversed movements more frequently in the rotational direction (Fig. 3c left and middle). The slopes of these plots for each longitudinal displacement give us the longitudinal velocities. The average longitudinal velocities of $M_2$ and $M_2C_2$ were 0.30 ± 0.03 μm s⁻¹ and 0.19 ± 0.05 μm s⁻¹, respectively (mean ± SD, $p < 0.001$, Kruskal-Wallis test with correction for multiple comparisons), (Fig. 3b left and middle). This reduction of the longitudinal velocity by CYK4 is consistent with the previously reported effect of CYK4 on the motility of the clustered MKLP1 along the surface-immobilized microtubules[3] and the gliding of the microtubules driven by the surface-immobilized MKLP1[3,16]. The average rotational velocities for $M_2$ and $M_2C_2$ were 0.37 ± 0.08 revolutions s⁻¹ and 0.12 ± 0.05 revolutions s⁻¹, respectively (mean ± SD, $p < 0.001$) (Fig. 3d left and middle). The bigger negative impact of CYK4 on the rotational velocity than on the longitudinal velocity accounts for the longer helical pitch of $M_2C_2$ (Fig. 2b left and middle). The torque generation by MKLP1 and its modulation by CYK4 were also observed in the microtubule gliding assay as well by 3D nanometer tracking of a quantum dot attached to the microtubule (Supplementary Fig. 3). This suggests that the bigger impact of CYK4-binding on the rotational motion than on the longitudinal one reflects the intrinsic properties of MKLP1 and CYK4 independent of the assay methods.

**Monomeric MKLP1, like heterotetrameric centralspindlin, showed a large fluctuation in the helical motion.** A recent structural study revealed that CYK4-binding restricts the configuration of the two MKLP1 motor domains in the centralspindlin complex and prevents the motor domains from simultaneously interacting with a single microtubule[3]. This explains the reduction of the longitudinal velocities by CYK4 (Fig. 3b and Table 1)[3,16] as monomeric constructs of kinesin motors usually showed slower microtubule gliding velocity than that of the dimeric constructs[36,37]. We hypothesized that the effect of CYK4 on the torque generation

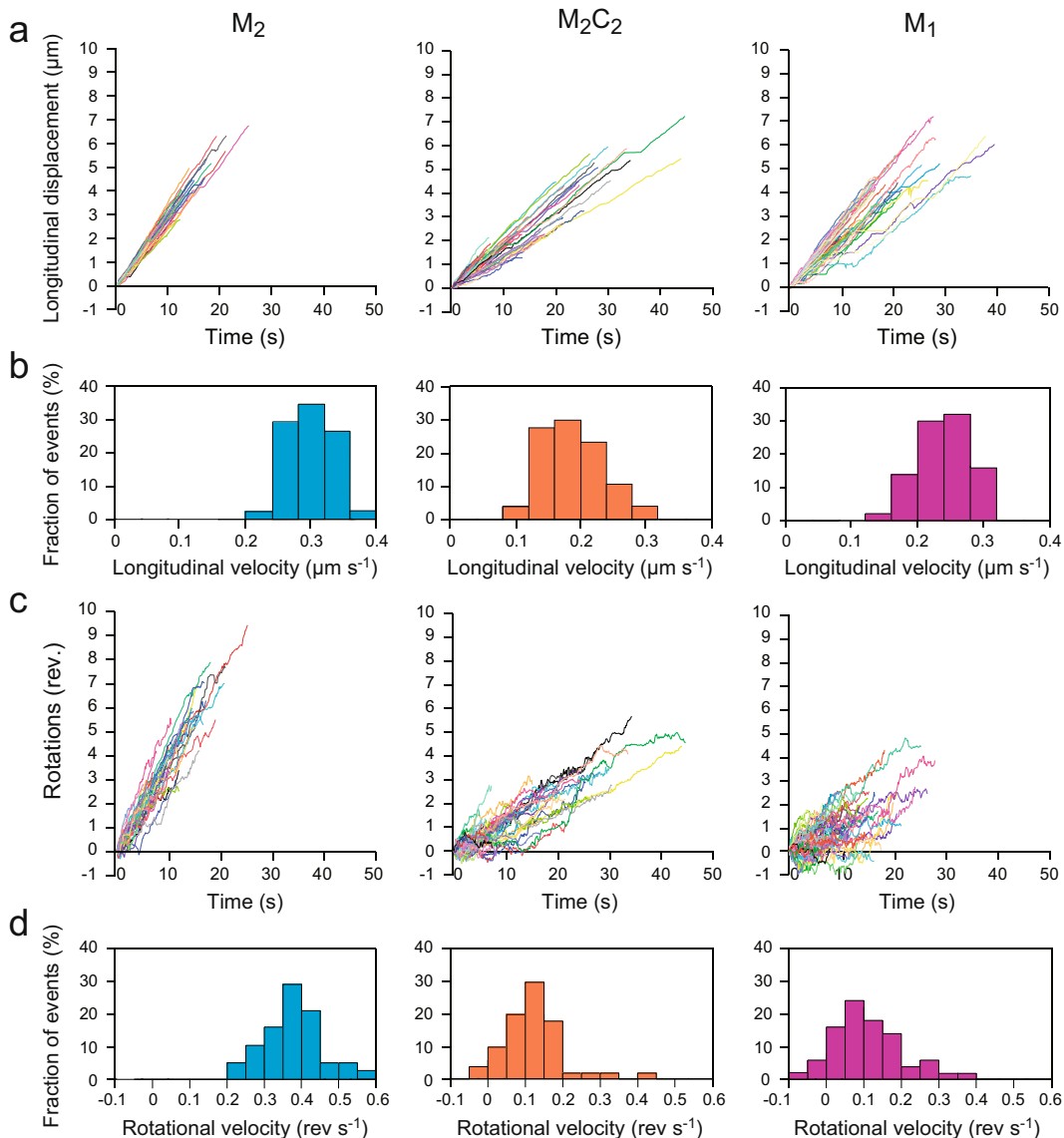

**Fig. 3 Time course of longitudinal and rotational motions by MKLP1. a** Time course of longitudinal distances of beads coated with M$_2$ ($n = 36$, left), M$_2$C$_2$ ($n = 47$, middle), M$_1$ ($n = 47$, right). **b** Histogram of the longitudinal velocity for M$_2$ (left), M$_2$C$_2$ (middle) and M$_1$ (right). Individual traces in **a** were fitted with linear functions to obtain the longitudinal velocity. Average longitudinal velocities, M$_2$: 0.30 ± 0.03 µm s$^{-1}$, M$_2$C$_2$: 0.19 ± 0.05 µm s$^{-1}$, M$_1$: 0.24 ± 0.04 µm s$^{-1}$. Data are given as mean ± standard deviation. **c** Time course of revolutions of beads coated with M$_2$ (left), M$_2$C$_2$ (middle), M$_1$ (right). **d** Histogram of the rotational velocity for M$_2$ (left), M$_2$C$_2$ (middle) and M$_1$ (right). Individual traces in **c** were fitted with linear functions to obtain the rotational velocity. Average rotational velocities, M$_2$: 0.37 ± 0.08 rev s$^{-1}$, M$_2$C$_2$: 0.12 ± 0.05 rev s$^{-1}$, M$_1$: 0.11 ± 0.09 rev s$^{-1}$. Data are given as mean ± SD.

by MKLP1 might also be caused by the prevention of the simultaneous interaction of the two motor domains. To test this idea, we observed the 3D movement of the beads coated with a monomeric MKLP1 construct, M$_1$ (Fig. 1a, b) along the suspended microtubule (Fig. 2a and Supplementary Fig. 2 right). Like the beads coated with M$_2$ and M$_2$C$_2$, the beads coated with multiple M$_1$ showed predominantly left-handed helical trajectories with an average longitudinal velocity of 0.24 ± 0.04 µm s$^{-1}$, an average rotational velocity of 0.11 ± 0.09 rev s$^{-1}$ and an average pitch of 2.1 ± 1.6 µm, respectively (mean ± SD) (Figs. 2b and 3 right). The M$_1$-driven helical motion, like the M$_2$C$_2$-driven one, showed large fluctuations with frequent pausing and occasional reversal in the rotational movement (Fig. 3c right). In rotational velocity and pitch, no significant differences were detected between M$_1$ and M$_2$C$_2$ ($p \sim 1$ and 0.37, respectively, Kruskal-Wallis test with correction for multiple comparisons) (Table 1). In contrast, the rotational velocity and pitch by M$_1$ were slower and longer than those by M$_2$ ($p < 0.001$ for

both) (Table 1). In a microtubule surface gliding assay with immobilized motors (Supplementary Fig. 3), M$_1$ also rotated the gliding microtubules. While the pitch generated by M$_1$ was also longer than that generated by M$_2$ ($p < 0.001$, Kruskal-Wallis test with correction for multiple comparisons), there was no significant difference between M$_1$ and M$_2$C$_2$ ($p = 0.30$) (Supplementary Fig. 3). Taken together, these observations suggest that the monomeric MKLP1 is more similar to the centralspindlin complex than to the dimeric MKLP1 in helical motions.

**Heterogeneity in the mode of motion.** The beads driven by the MKLP1 constructs showed helical motions with fluctuations both in longitudinal and rotational directions (Figs. 2 and 3), indicating that these motions are stochastic processes. This is consistent with the observed pitches of the helical motion, which correspond to one protofilament switch per around 7–20 forward steps on

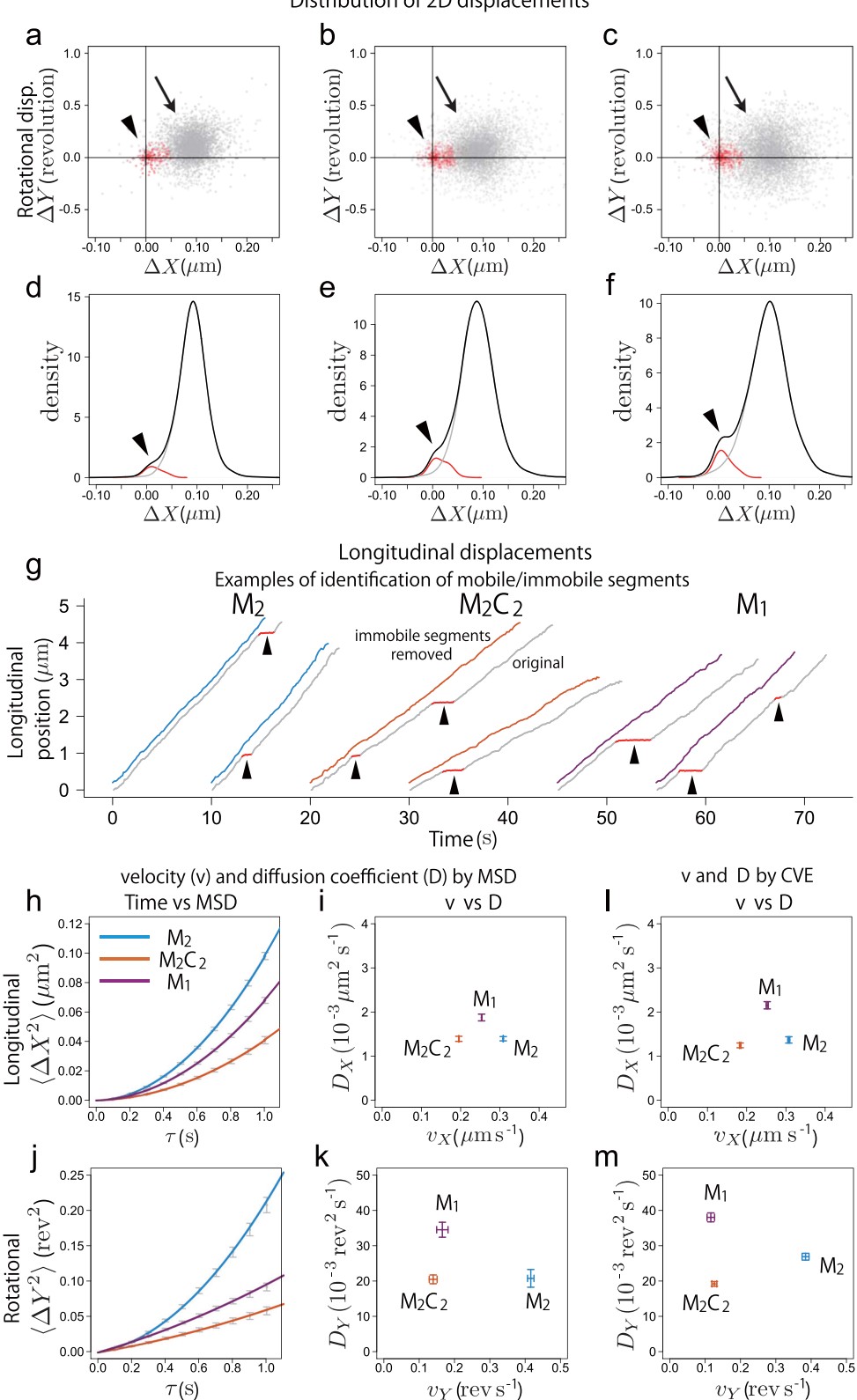

average. It is difficult to imagine a non-stochastic mechanism that can achieve such frequencies of off-axis movement. To analyze the stochastic nature of the motions driven by MKLP1 constructs, we first examined the homogeneity of the random process (Fig. 4a–g). If a trajectory is composed of random displacements caused by a single stochastic process, the displacements in a fixed time interval should exhibit a Gaussian distribution with a single peak.

However, the distributions of the longitudinal and rotational displacements exhibited two peaks for all three constructs (Fig. 4a–c). As expected from the left-handed helical motions, the major peaks (Fig. 4a–c, arrows) were slightly deviated off the longitudinal axis with positive rotations. In contrast, the minor peaks were found near the origin (Fig. 4a–c, arrowheads) and appeared as irregular 'shoulders' on major Gaussian distributions

**Fig. 4 Heterogeneities in the helical motions driven by the MKLP1 constructs and their velocities and diffusion coefficients for longitudinal and rotational motions.** **a–c** Distributions of the longitudinal ($\Delta X$) and rotational ($\Delta Y$) displacements along and around the microtubule indicate the heterogeneous modes in the motion of the beads. Displacements in the interval that roughly corresponds to the 0.1 µm movement along the microtubule (0.3 s, 0.5 s, and 0.4 s for $M_2$, $M_2C_2$, and $M_1$, respectively) showed distributions with two peaks, i.e., the major peak and the minor peak around the origin, which correspond to the mobile and immobile modes, respectively. The mode of each displacement was estimated by a Bayesian inference using a hidden Markov model that assumes that a bead switches between these unobservable modes. Red indicates that the displacement was inferred to be in the immobile mode. Gray indicates that the displacement was inferred to be in the mobile mode. **d–f** Probability densities of the longitudinal displacements ($\Delta X$) in the immobile (red) and mobile (gray) modes as well as the displacements in either mode (black) were calculated with the bandwidth of 0.01 µm. **g** Typical examples of the trajectories that were inferred to have an immobile segment (red). The segments inferred to be in the mobile mode (gray) were reassembled to generate a virtual trajectory consisted of the displacements uniformly in the mobile mode (blue: $M_2$, orange: $M_2C_2$, purple: $M_1$). **h–m** The data from the mobile segments were used to derive velocities and diffusion coefficients by mean square displacement (MSD) analysis (**h** to **k**) or by using the covariance-based estimator (CVE) (**l** and **m**). The mean square displacements of the longitudinal (**h**) or the rotational (**j**) motion in the indicated time intervals ($\tau$) were calculated for individual trajectories and averaged across the beads driven by the same construct (gray, mean ± SE, $n = 36$, 47 and 47 beads for $M_2$, $M_2C_2$, and $M_1$, respectively). They were fitted with a quadratic curve $\langle \Delta X^2 \rangle = y_0 + 2D_X\,\tau + v_X^2\tau^2$ or $\langle \Delta Y^2 \rangle = y_0 + 2D_Y\,\tau + v_Y^2\tau^2$ (blue: $M_2$, orange: $M_2C_2$, purple: $M_1$) ($v$: velocity, $D$: diffusion coefficient) and the determined parameters ($v$, $D$) were plotted (**i** and **k**, mean ± SE). The estimates of $y_0$ were 0 with standard errors smaller than 0.00002 µm$^2$ (**h**) and 0.0005 rev$^2$ (**j**). The velocity ($v$) for each trajectory was determined as the mean of the displacement ($\Delta X_i$) per observation interval ($\Delta t$), i.e., $v_X = \langle \Delta X_i \rangle / \Delta t$ and $v_Y = \langle \Delta Y_i \rangle / \Delta t$, and the diffusion coefficient ($D$) was calculated as $D_X = (\langle \Delta Z_i^2 \rangle / 2 + \langle Z_i\,Z_{i+1} \rangle) / \Delta t$ and $D_Y = (\langle \Delta W_i^2 \rangle / 2 + \langle W_i\,W_{i+1} \rangle) / \Delta t$ with the drift-adjusted displacements, $\Delta Z_i = \Delta X_i - v_X\Delta t$ and $\Delta W_i = \Delta Y_i - v_Y\Delta t$ for the longitudinal (**l**) and rotational (**m**) motions, respectively.

of the longitudinal displacements (Fig. 4d–f, black line, arrowheads).

To examine whether the minor component of the displacement distributions is scattered throughout a trajectory or clustered in a small number of places along the timeline, we carefully examined the trajectories. We found that the latter was the case, i.e., about a half of the trajectories (61 in 130 total) had a section in which the largely constant drifts in both the longitudinal and rotational coordinates temporarily halted and only a small random motion was observed (Fig. 4g for the longitudinal motion, sections marked with arrowheads on the original trajectories. Supplementary Fig. 4 for the rotational motion). After this confinement event, the beads restored the fluctuating helical motion with largely constant drifts similar to what they had previously exhibited. We couldn't detect any clear spatial or temporal pattern in the appearance of the confinement event. It seems as if, most of the time, a bead moves on the microtubule surface with a combined constant drift and random fluctuations, but is trapped at random sites for a while, although the causes for the entrapment and release remain unclear.

**Inference of the mode of motion with a hidden Markov model.** For proper characterization of the major mode of the motion, i.e, fluctuating helical drift, we need to separate the data representing this motion from the immobile section. To do this, we statistically modeled the series of displacements with a hidden Markov process that assumes that a bead stochastically switches between two modes of motion ('mobile' and 'immobile' modes) although the mode for each time point can't be directly observed. In the mobile mode, a bead follows a random fluctuation with a helical drift. In the 'immobile' mode, a bead is trapped on a site on the microtubule lattice. Detected changes of the bead position in this mode are attributed to the random fluctuation of the center of the bead relative to the microtubule-attachment site or the random noises and errors during image acquisition and analysis. Based on this model, the mode of motion for each time point could be estimated by Bayesian inference with the observed data without any prior knowledge about the hidden mode at individual time points. The result is presented in Fig. 4a–g by gray ('mobile' mode) and red ('immobile' mode) colors. The classification between the two modes seems reasonable. The beads were inferred to be in the 'immobile' mode only in a small fraction of the observed time points (5–10 %, Supplementary Table 1).

**Macroscopic parameters for longitudinal and rotational motions.** For the estimation of the velocity ($v$) and the diffusion coefficient ($D$), we pooled the displacements in the mobile mode and virtual trajectories consisted of the uniform random process were reconstituted (examples are shown by lines in blue ($M_2$), orange ($M_2C_2$) and purple ($M_1$) in Fig. 4g). Using these data, the mean square displacements along the longitudinal (Fig. 4h) and rotational (Fig. 4j) axes were calculated for the time intervals ($\tau$) from 0.1 s to 1.0 s (gray dots and error bars). Then, the MSD-interval data were fitted with a quadratic curve, MSD $= y_0 + 2D\tau + v^2\tau^2$ ($y_0$ is a constant) to estimate $v$ and $D$ (and $y_0$, although we obtained very small value for $y_0$), which are plotted in Fig. 4i, k.

Although MSD analysis has a long history, its disadvantages have recently become recognized. These include the lack of a simple guideline for the range of temporal intervals to be considered and the influences of the motion blur and the random noise in image acquisition and analysis on $y_0$[38–40]. To overcome these potential drawbacks, a novel method, termed covariance-based estimator (CVE) approach, has been developed to estimate the diffusion coefficient of an unbiased diffusion[41]. This method is free from fitting and only calculates a combination of the variance of the displacements in the minimum time interval (=the observation cycle) and the covariance between the displacements in the adjacent time points. It has been demonstrated that the CVE gives a more robust estimation of the diffusion coefficient than MSD analysis, especially for short trajectories[42]. We adapted the CVE approach to a biased diffusion by correcting the displacements for the constant drift (Fig. 4l, m). This approach gave us similar results to those by MSD analysis (Fig. 4h–k) although there are slight differences between them. The three constructs showed distinct longitudinal velocities while $M_2C_2$ and $M_1$ showed similar rotational velocities lower than that of $M_2$. On the other hand, as to the longitudinal and rotational diffusion coefficients, $M_2C_2$ showed similar values to those of $M_2$, lower than those of $M_1$. However, it is unclear how we should interpret these results.

**Modeling the stochastic helical motion as a biased random walk on the MT surface lattice.** To get better insights on the fluctuating helical motions of the beads driven by the MKLP1 constructs, we attempted to describe them as a 2D random walk, assuming that the bead moves by stochastic hopping to the nearby binding sites on the microtubule lattice surface at

anisotropic/biased rates. This is a mathematical simplification since we can't strictly exclude the possibilities against our assumption such as jumping beyond the neighboring sites. However, as shown below, comparison of the bead motion by different constructs on this scheme lead us to interesting results.

In the case of a one-dimensional, (biased) random walk, the stochastic rates of forward and backward hopping ($k_+$ and $k_-$, respectively) are linked to the velocity (or the rate of the constant drift) ($v$) and the diffusion coefficient ($D$) via equations, $v = k_+ - k_-$ and $2D = k_+ + k_-$. This means that the microscopic parameters such as the hopping rates can be deduced with the macroscopic parameters, $v$, and $D$, which can be determined with the observed trajectory data. Can't we determine the hopping rates on the 2D lattice in an analogous way? There are two potential obstacles. The first one is that, on the 2D lattice, there are 8 possible options for the hopping to the adjacent sites while the velocities and the diffusion coefficients along two axes can provide only 4 equations. The second one is that the geometry of the surface lattice of microtubules is not so simple due to a small shift (called 'rise') between the adjacent protofilaments, which result in the skew of the grid, the irregular contact (called 'seam') in one of the protofilament contacts, and the super-twist of the protofilaments around the axis of the microtubule when the protofilament number is not 13 (Fig. 5a). The first issue can be addressed by considering the covariance and higher-order covariances between the displacements in the 2 dimensions (see below). The second issue is more difficult to handle.

The skew of the lattice can be properly treated by assuming the number of protofilaments. Although we can't figure out the exact geometry of the individual microtubules, it is known that the majority of the in vitro-assembled microtubules used for our experiments have 13 or 14 protofilaments. They are more or less similar to each other and the analyses with the geometrical parameter sets for these configurations resulted in similar results (Supplementary Figs. 5 and 6). On the other hand, precise treatment of the seam is difficult and our model doesn't consider it. However, the influence of this is believed to be limited since the majority of the movement (12 in 13 or 13 in 14) occurs on the regular lattice. In addition, the covariance-based approach to calculate the macroscopic parameters such as the diffusion coefficient only uses the displacements in the minimum interval (the observation interval) and, thus, the effect of the seam remains local and doesn't propagate across the data through the calculation of the displacements of longer intervals.

## Inference of the stochastic hopping patterns on the MT lattice.
The diffusion coefficient of 1D motion is the coefficient for the temporal development of the variance of the displacement. Analogously, for a 2-dimensional motion, we can extend this to the temporal development of the co-variance and the higher-order co-variances. It can be shown that these (higher-order) co-variances are linear functions of time with a coefficient that can be expressed as a linear combination of the rates of hopping to the 8 adjacent sites. This means that we can deduce the microscopic parameters such as the hopping rates or the preferences of hopping direction from the macroscopic parameters ($v_z$, $v_w$, $D_z$, $D_w$, $A$, $B$, $C$, and $E$), which can be estimated from the trajectory data by an extended CVE-based approach. However, the simple solution of the linear relation sometimes results in negative values for the hopping rates, which should be zero or positive. This is likely due to the intrinsic stochasticity of the random walk and the sensitivity of the higher-order covariances to the extreme values, combined with the finite data size. This problem can be overcome by Bayesian inference with the assumption that the

observed macroscopic parameters are a random sample from the probabilistic distribution around the theoretical ones defined by the microscopic parameters, on which appropriate limitations ($k_i \geq 0$ and $0 \leq p_i \leq 1$) can be applied (Fig. 5b).

We started the analysis of the stochastic hopping on the MT lattice by converting the trajectory data from the mobile segments (Fig. 4) measured in longitudinal displacement and rotation ($X Y$) into the position on the lattice ($z w$) by the linear transformation that is defined in Fig. 5a, b assuming that the number of protofilaments $n = 13$ for Fig. 5 although very similar results were obtained with $n = 14$ as well (Supplementary Fig. 6). For each trajectory, $v_z$, and $v_w$, the rates of the constant drifts, were first calculated as the mean of the displacements in the regular observation interval (0.1 s). Then, the other macroscopic parameters ($2D_z$, $2D_w$, $A$, $B$, $C$, $E$) were estimated using the displacement data adjusted for the constant drifts (Supplementary Fig. 5). A set of these macroscopic parameters obtained for all the beads ($n = 36$, 47, and 47 for $M_2$, $M_2C_2$, and $M_1$, respectively) was used as the data for the Bayesian inference of the hopping rates ($k_1 \sim k_8$), which can also be expressed as $k_i = k \times p_i$, where $k = k_1 + \ldots + k_8$ and $p_1 \sim p_8$ are the hopping preferences ($p_1 + p_2 + \ldots + p_8 = 1$).

The posterior probability distributions of the microscopic parameters, i.e., the result of the inference, are shown in Fig. 5c. As expected, for all the constructs, the hopping occurred most frequently to the forward site on the same protofilament ($p_2$: ~0.8 for $M_2$ (blue) and ~0.6 for $M_2C_2$ (orange) and $M_1$ (purple)). Hopping to the other sites were also detected with lower probabilities. Hopping to the right-backward ($p_8$) showed the lowest probabilities in all the constructs. The patterns of the hopping preference are more graphically presented in Fig. 5d. We find an overall similarity between $M_2C_2$ and $M_1$, closer than between $M_2$ and them. $M_2$ is distinct especially in $p_2$, $p_3$, and $p_7$; higher $p_2$ (~0.8 for $M_2$ vs ~0.6 for $M_2C_2$ and $M_1$) and lower $p_3$ (<0.05 for $M_2$ vs 0.05~0.1 for $M_2C_2$ and $M_1$) and $p_7$ (<0.05 for $M_2$ vs 0.1~0.2 for $M_2C_2$ and $M_1$). In other words, while the motion of the bead driven by $M_2$ was largely restricted to the forward ($p_2$), left-forward ($p_1$), or left ($p_4$), those by $M_2C_2$ and $M_1$ was more relaxed, exemplified by the higher probabilities towards the right-forward ($p_3$) and the straight-backward ($p_7$). The biggest difference between the heterotetramer ($M_2C_2$) and the monomer ($M_1$) is in the overall rates of transition, $k$. Essentially the same results were obtained with coordinate conversion assuming the 14-protofilament configuration (Supplementary Fig. 6).

In conclusion, our data indicate that the stochastic hopping driven by the dimeric MKLP1 ($M_2$) is more tightly restricted towards the forward, forward-left and left directions than that by the monomeric one ($M_1$) and that the binding of the CYK4 dimer to the neck of the dimeric MKLP1 relaxes the directional restriction of the dimeric construct, making the heterotetramer ($M_2C_2$) more like the monomer ($M_1$).

## Correlation between the affinity to microtubules during the ATPase cycle and the bias in the direction of the stochastic motion.
To get further insights into the mechanisms behind the biased protofilament switching, we analyzed the ATPase activity of the MKLP1 constructs in solution that is activated by microtubules (Supplementary Fig. 7). As summarized in Table 1, they showed similar $k_{cat}$ values, which represent the maximum rates of the enzymatic cycle that they can achieve. By contrast, major differences were found in their Michaelis constants ($K_M$), indicating that they interact with microtubules with different affinities during the ATPase cycle. The order in their $K_M$ values is consistent with that of the minimum motor:bead ratio necessary for the continuous bead motion along the microtubule (for example, $M_2$, the construct

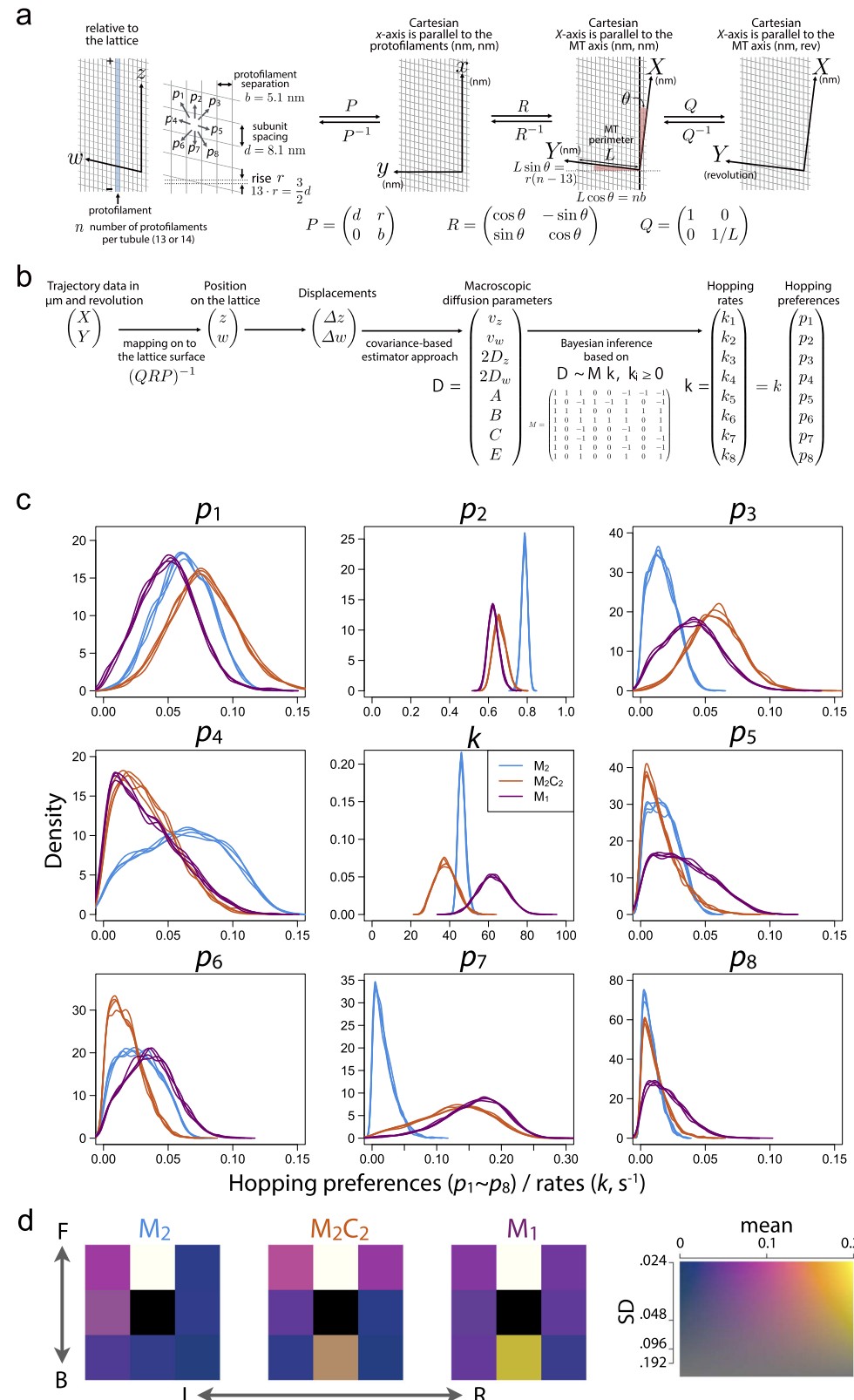

with the lowest $K_M$, could support the continuous bead motion with the lowest motor:bead ratio). The order in $K_M$ values also correlated with the extent of the bias in the direction of the stochastic motion of the bead driven by MKLP1 (Fig. 5). $M_2$, which showed the highest affinity to microtubules, exhibited the strongest bias in the off-axis motion while $M_1$ with the lowest affinity showed the mildest bias, and $M_2C_2$ was intermediate in both the activities. Previous works suggested that CYK4-binding to the neck of the MKLP1 dimer prevents the simultaneous binding of the two motor domains to the same microtubule[3]. This might provide a common explanation both for centralspindlin heterotetramer's weaker affinity to the microtubule during the ATPase cycle and for its higher flexibility in the direction of the stochastic motion as a team of motors than dimeric MKLP1.

**Fig. 5 Inference of the directional preferences of the stochastic motion by MKLP1 on the microtubule lattice surface. a** Interconversion between the coordinate systems. A microtubule consists of 13 or 14 protofilaments ($n = 13$ or $14$), linear assembly of the heterodimers of alpha/beta-tubulin with $d = 8$ nm spacing, aligned in parallel with a small shift in the adjacent alpha/beta-dimers ('rise' $r = 0.82$ nm)[65]. As the sum of the 13 rises is 12 nm = $1.5 \times 8$ nm, a microtubule with 13 protofilaments has a seam, a protofilament interface with a mismatch in alpha/beta subunit alignment, but the protofilaments are parallel to the axis of the tubular assembly. The sum of the 14 rises is not a half-integer multiple of 8 nm spacing. This results in a small tilt (0.73°) of the protofilaments against the microtubule axis, theta. The position along the $Y$-axis perpendicular to the $X$-axis parallel to the microtubule axis can be converted to the revolution via the length of the perimeter $L$. Thus, the mismatch at the seam being put aside, the trajectory data ($X$ $Y$) measured by nm and revolution can be linked to the position on the lattice ($z$ $w$) as ($X$ $Y$) = $Q$ $R$ $P$ ($z$ $w$). **b** Workflow to infer the hopping rates and preferences. The trajectory data from the mobile segments (Fig. 4) were mapped onto the lattice via the linear transformation ($Q$ $R$ $P$)$^{-1}$. The macroscopic parameters ($v$, $D$, and the coefficients for the higher-order covariances, $A$, $B$, $C$, $E$) were calculated and used to infer the microscopic parameters, i.e., the hopping rates and preferences based on the linear relationship between them. **c, d** The results of the Bayesian inference of the hopping rates and preferences assuming $n = 13$ (Similar results were obtained with $n = 14$, See Supplementary Fig. 6). The posterior probability distributions of the overall hopping rate ($k$) and preferences ($p_1 \sim p_8$, see **a**) (**c**) were presented as the density distributions of the four chains of sampling for each construct. The means and standard deviations of the hopping preferences (**d**) were graphically represented with a color scale shown on the right. The strong bias towards left and forward observed in $M_2$ was lost in $M_2C_2$ and $M_1$ (compare them on $p_3$ (forward-right) and $p_7$ (straight backward)). Note that the probability of hopping to the straight forward site ($p_2$) is much higher than other probabilities and thus it appears saturated (white).

## CYK4-binding helps MKLP1 avoid getting stuck by obstacles.

How can flexible off-axis motion of centralspindlin be beneficial in a biological context? Timely accumulation to the central microtubule overlaps of the central spindle is crucial for the functions of centralspindlin as a key microtubule organizer and a signaling hub for cytokinesis. To achieve this in the crowded environment of the spindle midzone, centralspindlin needs to move along microtubules avoiding obstacles such as other kinesins and microtubule-associated proteins (MAPs). Flexible choice in the direction of the stochastic motion promoted by CYK4-binding might facilitate the bypassing of obstacles through protofilament switching, which has been suggested by previous in vivo and in vitro assays using kinesin-1[43], kinesin-2[44], and kinesin-8[45]. To test this, we tracked 3D displacements of $M_2$-coated and $M_2C_2$-coated beads along a microtubule immobilized on the coverglass surface (Fig. 6a), instead of being suspended away from the surface. Before encountering the glass surface, the $M_2$-coated beads showed both motions along and around the surface-immobilized microtubule. However, afterwards, the motion was confined along the longitudinal axis, keeping contact with the glass surface barrier, which prevented $M_2$ from taking further leftward motion (Fig. 6b, c). By contrast, $M_2C_2$-coated beads were able to continue off-axis movement by reverting rotational direction after contacting the glass surface barrier (Fig. 6d, e). Importantly, while the longitudinal velocity of $M_2$-coated beads on the surface-immobilized microtubules ($0.21 \pm 0.04\ \mu m\ s^{-1}$ (mean ± SD)) was significantly lower than that on the suspended microtubules ($0.30 \pm 0.03\ \mu m\ s^{-1}$ (mean ± SD)), ($p < 0.001$ with Wilcoxon rank-sum test), that of $M_2C_2$-coated beads did not show such a reduction even in the presence of the surface as an obstacle ($0.19 \pm 0.05\ \mu m\ s^{-1}$ vs $0.15 \pm 0.05\ \mu m\ s^{-1}$ (mean ± SD), $p \sim 0.1$ with Wilcoxon rank-sum test) (Table 1). This suggests that CYK4-binding aids centralspindlin in avoiding obstacles by enabling more flexible choice of stepping directions than dimeric MKLP1 alone.

## Discussion

In contrast to the canonical N-kinesins, the force-generating mechanisms of the non-canonical N-kinesins, which lack a typical neck linker but instead have a long neck, have been poorly studied. Here we examined the motion of a bead driven by a non-canonical N-kinesin, MKLP1 kinesin-6, around a freely suspended microtubule by 3D-nanometry using the *tPOT* microscope. We observed the torque-generation by a non-canonical N-kinesin. MKLP1-coated beads exhibited left-handed helical trajectories with some fluctuations around the microtubule lattice surface. The

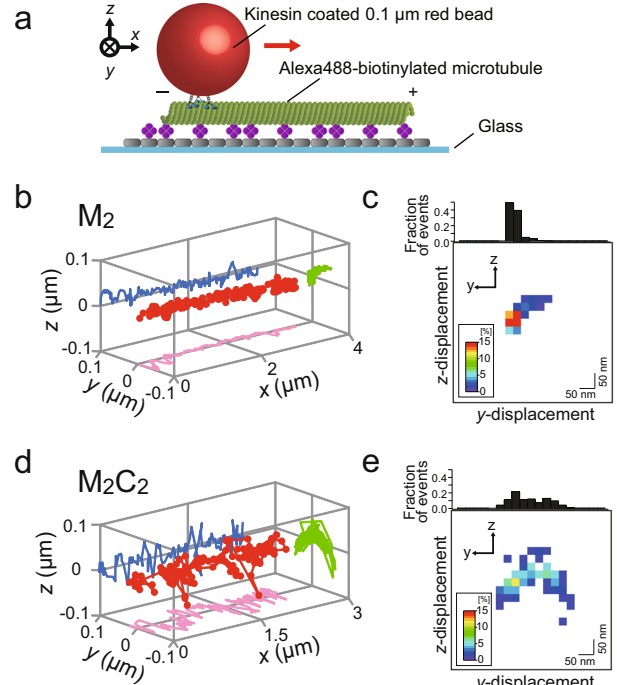

**Fig. 6 Continuous rotational movement displayed by $M_2C_2$ in a spatially restricted environment. a** Schematic representation of the experimental geometry. The biotinylated, Alexa488-labeled taxol stabilized microtubule was attached to the biotinylated BSA (gray)-coated coverglass via streptavidin (purple). Using tPOT, the MKLP1-coated bead movements on the non-suspended microtubule were quantified. **b** A 3D trace of an $M_2$-coated bead moving along the coverglass surface-immobilized microtubule. Initially, the $M_2$-coated bead showed a left-handed rotation. After reaching the left edge of the microtubule and contacting the glass surface, the $M_2$-coated bead suspended the rotational motion although it continued the longitudinal motion. **c** Bivariate histogram showing normalized distribution of the $y$–$z$ location of the bead shown in **b**. The colors indicate the frequency of finding the bead in each 25 nm square on the $y$–$z$ projection. $M_2$ showed limited rotational motion on the coverglass surface-immobilized microtubule. **d** A 3D trace of $M_2C_2$-coated bead along a microtubule immobilized on the coverglass surface. Upon reaching the left edge of the microtubule and contacting the glass surface, the bead often reversed rotational direction and continued the rotational motion. **e** Bivariate histogram showing normalized distribution of the $y$–$z$ location of the bead shown in **d**.

pitch and persistence of the trajectories were modulated by the dimeric state and the binding of CYK4 to the atypically long neck region of MKLP1, which forms the stable centralspindlin complex crucial for cytokinesis. CYK4 also changed the response of the MKLP1-driven motion in the protofilament switching upon an encounter with an obstacle.

It has been demonstrated that monomeric kinesins-1, -3, and -5 generate a left-handed torque on the surface gliding assays[21,24,27]. We have found that monomeric MKLP1 kinesin-6 also drives a left-handed helical motion of a bead around a microtubule albeit in a longer pitch and with more fluctuation than the dimeric one. These indicate that the left-handed torque generation is a common, intrinsic characteristic of the monomeric kinesin motor domain. Although we can't exclude the possibility of mechanical communication between multiple motors on a bead, this effect alone would not be able to explain the torque generation since the arrangement of the motors on a bead or coverglass surface should be random. The torque generation by the monomeric kinesin motor domain must be accounted for by some structural bias in the mechanical cycle, which would underlie the asymmetric potential landscape in the context of the Brownian ratchet mechanism[27,46]. In analogy with the proposed mechanisms for the motion along the microtubule axis by monomeric kinesins[47], the candidate mechanisms would include the conformational changes in the catalytic motor core or in the neck region[2], and electrostatic guidance[48] upon landing on or detachment from the microtubule. However, the exact mechanisms remain unclear.

Although dimeric kinesin-1 follows a single protofilament, elongation of the neck-linker, like the monomerization discussed above, makes it generate a torque both in the assay with a bead on a suspended microtubule[19] and in the surface gliding assay[20]. The sequence in the neck region affects the helical pitch of the bead coated with kinesin-2[19]. Here we have demonstrated that dimeric MKLP1 kinesin-6 exhibits a helical trajectory with a shorter pitch than the monomeric one and that CYK4-binding to the long neck of the MKLP1 dimer partially cancels out the effect of dimerization. Collectively, these indicate that the torque generation by kinesins is modulated by the coordination or communication between the two heads in a dimer, although the exact details of the modulation vary widely depending on the types of kinesins. Again, the mechanisms proposed for the head-to-head communication in the coordinated motion of dimeric kinesin-1 along the microtubule axis might explain the modulation of the torque as well. These include the diffusive search of the leading head[20,24,27,28] and anisotropic detachment of the trailing head via intramolecular strain[49–51].

We have found that centralspindlin (heterotetramer of MKLP1 and CYK4) behaves more like monomeric MKLP1 than dimeric MKLP1 in the microtubule-activated ATPase and in the directional preference of the stochastic helical motion around a microtubule. Our previous work revealed that CYK4 binding to the neck of MKLP1 makes the configuration of the two motor domains in a dimer less flexible and thus prevents their simultaneous binding to the same microtubule[3]. We speculate that the presence of another motor head flexibly linked via long necks might facilitate the leftward torque generated by a motor head to be translated into the proto-filament switching towards the left. CYK4-binding might interfere with this head-to-head communication in the dimer and thus relaxes the bias in the protofilament switching although we should be cautious about linking our observation to a structural mechanism since multiple motors randomly attached on a bead are involved in the helical motion of a bead in our experiments.

We don't know why the MKLP1-driven bead occasionally shows a temporary pause during its helical motion around the microtubule. This might be due to some irregularities in the lattice structure such as the seam or random lattice defects[34]. Some tubulin isoforms or some post-translational modifications of tubulin might also interfere with the bead motion driven by MKLP1. MKLP1 dimer and the heterodimer with CYK4 shows strong accumulation to the plus-end of the microtubule both in vitro and in vivo[3,14,52–54]. The temporary pause observed in our in vitro assay might be reflecting an unknown property of MKLP1 kinesin-6 that promotes the plus-end accumulation.

To characterize the fluctuating helical motion on the microtubule lattice surface, we have developed a novel method to estimate the microscopic parameters of the 2-dimensional stochastic hopping based on the macroscopic parameters such as the coefficients for the time-development of the (higher-order) covariances between the displacements along the two axes. In this, the CVE-based approach extended to the higher-order covariances played an important role. Although we assumed that the bead moves on the microtubule lattice by hopping to the neighboring sites, we need to keep in mind that this a mathematical simplification and might not strictly be the case. For example, the bead might jump beyond the neighboring sites by rapid detachment and attachment. Nonetheless, by comparing the bead motion by different constructs with our new method, we could obtain important insights on the hopping preferences, which could not be achieved just by determining the longitudinal and rotational velocities and diffusion coefficients. We believe that this method can be used to analyze fluctuating helical motions of other motor proteins around microtubules.

Currently, the biological significance of the helical motion or the torque-generation by MKLP1 kinesin-6 remains unclear. Centralspindlin is a key microtubule organizer and signaling hub for cytokinesis. For these functions, timely accumulation of centralspindlin to the equatorial zone of anti-parallel overlaps of the central spindle microtubules is crucial. During the travel towards the plus-end of a microtubule, dimeric MKLP1, which has a strongly biased directional preference (Fig. 5), might be entrapped when it encountered obstacles that block the forward and left motions. By contrast, centralspindlin heterotetramer would be able to bypass such obstacles by moving backwards or by proceeding towards the right-front. We observed that, on the microtubule immobilized on a surface, the motion of a bead by MKLP1 dimer was confined to the left side of the microtubule while that by the MKLP1-CYK4 heterotetramer kept switching the protofilaments (Fig. 6). Thus, the ability of flexible proto-filament switching in a confined environment might help the rapid transport of centralspindlin in the highly crowded spindle environment without full detachment from a microtubule, which would make the transport less efficient. An alternative, but not exclusive, possibility would be the relaxation of the torque applied to the spindle. A torque-generating and crosslinking minus-end directed kinesin, kinesin-14, slides parallel and anti-parallel microtubule overlaps in different ways and induces a twist in a locked anti-parallel bundle[55,56]. The torque generation, in addition to the optimized head configuration by CYK4[3], might contribute to preferred accumulation of centralspindlin to the anti-parallel overlaps pre-templated before anaphase onset. Torque by kinesin-5, a plus-end directed motor and crosslinker, introduces a helical twist onto the microtubule organization of the mitotic spindle[57]. CYK4-binding, which relaxes the directional preferences of the MKLP1 dimer, might contribute to alleviating the helical twists of the central spindle and the midbody. Future studies with higher spatial and temporal observations of the in vivo dynamics of centralspindlin and the microtubules would be necessary to reveal the biological roles of stochastic proto-filament switching.

## Methods

**Construction of plasmids.** The construction of the GST-CYK-4(1–360 nt)/pGEX-6p plasmid has been described[3]. To generate the $M_2$ (ZEN-4, 1–1665 nt)-SNAP-His/pET32a plasmid, an *Eco*RI fragment (1334–1665 nt) of CBD-TEV-ZEN4 (1–1665 nt)/pGEX-6p[3] was inserted into SNAP-His/pET32a plasmid. The plasmid was digested with *Nde*I and *Hind*III, and the *Nde*I-*Hind*III fragment (1–1624 nt) of CBD-TEV-ZEN4/pGEX-6p was inserted. To generate the $M_1$ (ZEN-4, 1–1317 nt)-SNAP-His/pET32a plasmid, the PCR-amplified *Nde*I-*Hind*III fragment (1–1317 nt) of CBD-TEV-ZEN4 (1–1665 nt)/pGEX-6p, which removed the internal *Eco*RI restriction site (312 nt) by site-directed mutagenesis, was digested with *Nde*I and *Hind*III and was inserted into the *Nde*I and *Hind*III sites of SNAP-His/pET32a plasmid digested by the same enzymes. All constructs were confirmed by DNA sequencing.

**Expression and purification of proteins.** For expression and purification of both $M_2$ and $M_1$, C-terminally SNAP-his tagged proteins were expressed in *Escherichia coli* strain BL21 Star (DE3) cells using 0.4 mM IPTG (Sigma-Aldrich) for 9~12 h at 20 °C. The cells were collected and were lysed in lysis buffer (100 mM Tris; pH 8.0, 50 mM KCl, 1 mM MgCl$_2$, 100 μM ATP, 20 mM β-mercaptoethanol, 0.1% CHAPS, 0.1% Tween20, 10% glycerol, protease 0.1 mg ml$^{-1}$ lysozyme, protease inhibitors) and sonicated on ice for 12 min. The lysates were clarified by ultracentrifugation. The proteins were purified by affinity chromatography using an ÄKTA system equipped with a HisTrap HP Ni$^{2+}$-Sepharose column (GE Healthcare). The collected peak fraction was reapplied to a HiTrap desalting column (GE Healthcare) to exchange to the B buffer (20 mM potassium phosphate; pH 7.4, 1 mM MgCl$_2$, 5 mM β-mercaptoethanol, 200 μM ATP) containing 80 mM NaCl (for $M_1$) or 250 mM NaCl (for $M_2$). The purified proteins were flash frozen and stored in liquid nitrogen. For expression and co-purification of $M_2C_2$, both $M_2$-SNAP-His and GST-$C_2$ were expressed in *Escherichia coli* strain BL21 Star (DE3) cells using 0.2 mM IPTG for 8 h at 25 °C. The cells were collected by centrifugation. Those cells were lysed in lysis buffer (100 mM Tris; pH 8.0, 50 mM KCl, 1 mM MgCl$_2$, 100 μM ATP, 20 mM β-mercaptoethanol, 0.1% CHAPS, 0.1% Tween20, 10% glycerol, protease 0.5 mg ml$^{-1}$ lysozyme, protease inhibitors) and mixed. The lysate was sonicated for 25 min on ice and was clarified by centrifugation. The sonication supernatant was incubated on ice for 30 min to induce the interaction between $M_2$ and $C_2$[58]. The complex was purified by tandem affinity chromatography using Ni Sepharose 6 Fast Flow beads (GE Healthcare) and glutathione-Sepharose 4B beads (GE Healthcare) in batch operation. The N-terminally GST tag of $C_2$ was cleaved using Precision protease (GE Healthcare) during tandem purification. The complex was further purified by gel filtration using an ÄKTA system equipped with a Superdex 200 gel filtration column (GE Healthcare) and exchanged in the B buffer containing 80 mM NaCl. The purified proteins were flash frozen and stored in liquid nitrogen. The concentrations of purified proteins were estimated by SDS-PAGE on 10% acrylamide gels using BSA standards (Thermo fisher scientific) loaded on the same gel[59]. Gels were stained with Quick-CBB PLUS (Wako) and imaged using a CCD camera (CSFX36BC3, Toshiba). The bands containing purified proteins and BSA standards were quantified using ImageJ (NIH).

**Microtubule preparation.** Tubulin was purified from porcine brains through four cycles of temperature regulated polymerization and depolymerization in a high molarity PIPES buffer to remove contaminating MAPs[60]. The purified tubulin was flash frozen and stored in liquid nitrogen. Biotinylated (biotin-(AC5)2-Sulfo-OSu, Dojindo), Alexa 488-labeled (Alexa Fluor 488 Succinimidyl Ester, Thermo Fisher Scientific) microtubules were prepared by co-polymerizing biotinylated, Alexa488-labeled and non-fluorescent tubulin in a molar ratio of 1:5:170 in BRB80 (80 mM PIPES; pH 6.8, 1 mM MgCl$_2$, 1 mM EGTA) with 1 mM MgCl$_2$ and 1 mM GTP for 30 min at 37 °C then stabilized by addition of 40 μM taxol (Sigma-Aldrich)[21].

**Fabrication of microstructures on glass using plasma etching.** As illustrated in Fig. 1c, microgrooves are fabricated using the standard photolithography technique[61]. Briefly, SU-8 photoresist (SU-8 5, MicroChem, MA, USA) was spin-coated onto a cleaned glass substrate (150 μm in thickness; Matsunami Glass Ind., Japan) and exposed to UV radiation (100 mJ cm$^{-2}$) through a photomask. After a development process, the unmasked part of the glass substrate was etched to form grooves of 2 μm depth and 10 μm width using a neutral loop discharge (NLD) dry etching apparatus (NLD-5700Si, ULVAC, Japan, etching gas: O$_2$/He/Ar/CHF$_3$/ C$_4$F$_8$ (10/10/270/10/10 sccm); Chamber pressure: 0.67 Pa; Antenna power: 1500–1800 W). Finally, the SU-8 photoresist mask was removed from the patterned glass substrate using a piranha solution.

**3D prismatic optical tracking microscope.** The 3D prismatic optical tracking (termed *tPOT*) microscope was constructed as described previously[21,22]. As illustrated in Fig. 1d, the back-focal-plane (BFP) of the objective was focused outside the camera port of an inverted microscope (the dotted-line box, TE2000-U, Nikon, Tokyo, Japan) with achromatic Lens1 (combined focal length 50 mm) to make an equivalent BFP (eBFP). To split the image beam path at the eBFP, a custom-made wedge prism (86.5°) coated with an antireflective layer was precisely located at the eBFP. The two split images of a sample were focused on the camera focal plane by achromatic Lens2 (combined focal length 120 mm). Images were recorded by an

EM-CCD camera (iXon X3 DU897, Andor Technology, Belfast, UK) via Solis software (Andor) on a Windows 7 PC with 100 ms exposure every 100.28 ms for a typical 0.1 s imaging cycle. The fluorescently labeled microtubule, QDs and carboxylate-modified bead were illuminated using a mercury lump (Intensilight, Nikon, Tokyo, Japan). For calibration of the *z*-axis position the objective was displaced at a constant speed using a custom-built stable stage (Chuukoush-aSeisakujo, Tokyo, Japan) equipped with a pulse motor (SGSP-13ACT, Sigma Koki, Tokyo, Japan) and controller (QT-CM2, Chuo Precision Industrial, Tokyo, Japan). To suppress positional drift, which is mainly caused by temperature variations, a temperature-control unit (F-12, Julabo, Germany) was integrated into the *tPOT*. The temperature of the sample is maintained at 23.0 ± 0.2 °C (mean ± SD), which was measured by a micro thermo sensor (HD-1100E, Anritsu Meter, Tokyo, Japan).

**Preparation of kinesins-coated bead solutions.** The SNAP-tagged motor constructs were immobilized onto 0.22 μm diameter carboxylate-modified polystyrene beads (Thermo Fisher Scientific) via an anti-SNAP antibody (NEB). First, the antibody was mixed with the beads and adsorbed to the bead surface via non-specific binding. After 30 min incubation at room temperature, 0.2 mg ml$^{-1}$ casein was added to prevent further non-specific binding and the beads were washed with M buffer (20 mM PIPES, 10 mM potassium acetate, 4 mM MgCl$_2$, 1 mM EGTA, pH7.4) to remove both the remaining anti-SNAP antibody and casein. Then, the antibody-coated beads were incubated for 30 min at room temperature with 2.7 μM $M_2$, 28.5 μM $M_1$ or 4.1 μM $M_2C_2$ diluted in the M buffer containing 80 mM NaCl (4 μl). The beads collected by centrifugation were re-suspended into the M buffer (100 μl) supplemented with 0.5 mg ml$^{-1}$ casein, 2.7 mM ATP, 20 μM taxol, 0.5% β-mercaptoethanol, ATP regeneration and oxygen scavenger system, and made monodispersed by brief sonication. The re-suspended beads were observed within ~25 min.

**3D-motility of a motor-coated bead around a suspended microtubule.** A flow chamber was assembled with 20 mm × 30 mm patterned glass coverslip (150 μm in thickness; Matsunami Glass Ind., Japan) and 18 mm × 18 mm unpatterned glass coverslips (130–170 μm in thickness; Matsunami Glass Ind., Japan) using two thin strips of tape (810-1-18, Scotch 3 M) placed with a 5 mm gap as spacers inside the two parallel lines of grease. To suspend microtubules away from the glass surface, one of the coverslips was patterned with parallel walls as described above. The flow chamber was incubated with 1 chamber volume of 5 mg ml$^{-1}$ biotinylated-BSA (Sigma-Aldrich) in M buffer for 5 min, washed with 4 chamber volumes of cold M buffer, and incubated for 5 min with 1 chamber volume of 1 mg ml$^{-1}$ streptavidin (Sigma-Aldrich) in M buffer at room temperature. Following a wash with 4 chamber volumes of W buffer (0.5 mg ml$^{-1}$ casein and 10 μM taxol in M buffer), Alexa488-biotin-labeled microtubules were loaded as a suspension in 1 chamber volume of the same buffer and incubated for 7 min at room temperature. The non-immobilized microtubules were removed by a wash with 4 chamber volumes of the W buffer. Finally, the chamber was loaded with 2 chamber volumes of kinesin-coated bead solution, sealed with grease, and mounted on a *tPOT* microscope to measure the bead motion at 23 °C[21]. Beads on a microtubule suspended straight were chosen and recorded every 0.1 s with the *tPOT* system to track their 3D positions (*x*: parallel to the longitudinal axis, *z*: perpendicular to the imaging plane, *y*: perpendicular to both *x* and *z*). The rotational angle, θ, was calculated from (*y*, *z*). We analyzed the beads that showed a stable motion (> 1 μm) along a stably suspended single microtubule. Those on fluctuating, bundled, or crossed microtubules, and those on a non-suspended part of a microtubule, i.e., the segment bound on the patterned hilltop, were ignored. The beads that encountered other ones and those that showed a helical trajectory whose radius was too large by some unknown reason (> 250 nm ~ twice as large as the sum of the radii of the microtubule and the bead) were also excluded. Finally, we obtained 36 $M_2$, and, 47 $M_2C_2$, and 47 $M_1$ trajectories. All of these showed rotational motions around the microtubule although 13 $M_2C_2$ and 6 $M_1$ trajectories were too short to complete one full rotation. Individual longitudinal and rotational velocities were determined by fitting the *t-x* and *t-rev* trajectories, respectively, with linear functions. The inverse of the pitch was determined by fitting the *x-rev* trajectories with a linear function and dataset was averaged. Both the helical twist of the protofilaments (8 μm pitch ($P_{MT}$) for the 14-protofilament microtubule, none for the 13-protofilament microtubule) and the helical motion of the bead relative to the microtubule lattice surface (with a corrected pitch, $P_{motor}$) contribute to the observed helical pitch ($P_{obs}$). With the well-known relation between them, $1/P_{obs} = 1/P_{motor} + 1/P_{MT}$[19,33,62], and the proportion of the 13-protofilament microtubules in the mixture with the 14-protofilament microtubules of our preparation (α = 5/8)[21], the corrected pitch of the motor is estimated to be $P_{motor} = \frac{P_{obs} - P_{MT} + \sqrt{(P_{obs} - P_{MT})^2 + 4\alpha P_{obs} P_{MT}}}{2\alpha}$. However, with the actual values of $P_{obs}$ and α, its difference from the observed pitch $P_{obs}$ is less than 10%. Thus, we report the values of the observed pitch without correction. Statistical significance was analyzed by the Kruskal-Wallis rank sum test followed by post-hoc test after Dunn with *p*-value adjustment for multiple comparisons.

**Analysis of the trajectories of the bead motion.** Inference of the mode of motion was performed by a Bayesian inference, assuming that a bead switches between the two modes that cannot be directly observed: the mobile mode, in which a bead

exhibits a 2-deminensional random walk, and the immobile mode, in which a bead is trapped on a site on the MT surface. In both modes, displacements in a sufficiently long time are expected to follow a two-dimensional Gaussian distribution since a sum of a large number of repeated binomial processes, fluctuation of the bead around the contact site, and the localization errors during image acquisition and analysis are all approximated by a Gaussian distribution. The distances moved in a fixed time interval from each time point were calculated using the $X$- and $Y$-displacements normalized with the standard deviations, and used as data for a hidden Markov model implemented with Stan (http://mc-stan.org/) on R (https://www.R-project.org/)[63] to infer the hidden mode of motion. Separation between the two peaks corresponding to the mobile and immobile modes is improved by increasing the interval for measuring the displacements though at the sacrifice of the temporal resolution. A manual search (data not shown) for an optimal way of measuring the displacements that is commonly applicable to the different constructs revealed that the expected time for about 0.1 μm movement along the microtubule axis (0.3 s, 0.5 s, and 0.4 s for $M_2$, $M_2C_2$ and $M_1$ beads, respectively) gives a reasonable balance between the separation of the peaks and the temporal resolution.

To determine the velocity and the diffusion coefficient by analyzing the mean square displacements (MSD), for each dimension, the displacements in an interval from 0.1 s to 1.0 s were measured in all the possible combinations in a trajectory removed of the immobile segments, and the MSDs calculated for each trajectory were averaged across the trajectories to obtain the mean and standard error for each time interval. The mean MSD vs time interval data were fitted by a quadratic curve with the inverse variance weighting using the "port" algorithm by the nls() function on R with lower bounds of 0 for the parameters $y_0$, $v$, and $D$.

For determination of the velocity and the diffusion coefficient by the covariance-based estimator (CVE) approach, the displacements in the observation interval (0.1 s) were measured from the mobile segments of all the trajectories of the same construct and pooled. The velocity was calculated as the mean displacements per observation interval. The diffusion coefficient was calculated by applying the drift-adjusted displacements to the formula for the CVE of unbiased diffusion.

**Inference of the hopping rates and preference**. The inference of the rates and preferences of the hopping to the 8 surrounding sites was performed as described[64]. It is based on the linear relationship between these microscopic parameters and the 8 macroscopic parameters, i.e., velocities and diffusion coefficients for $x$- and $y$-axes as well as the coefficients for temporal development of the (higher-order) covariances between the $x$- and $y$-displacements.

$$
\begin{pmatrix} v_x \\ v_y \\ 2D_x \\ 2D_y \\ A \\ B \\ C \\ E \end{pmatrix} = \begin{pmatrix} 1 & 1 & 1 & 0 & 0 & -1 & -1 & -1 \\ 1 & 0 & -1 & 1 & -1 & 1 & 0 & -1 \\ 1 & 1 & 1 & 0 & 0 & 1 & 1 & 1 \\ 1 & 0 & 1 & 1 & 1 & 1 & 0 & 1 \\ 1 & 0 & -1 & 0 & 0 & -1 & 0 & 1 \\ 1 & 0 & -1 & 0 & 0 & 1 & 0 & -1 \\ 1 & 0 & 1 & 0 & 0 & -1 & 0 & -1 \\ 1 & 0 & 1 & 0 & 0 & 1 & 0 & 1 \end{pmatrix} \begin{pmatrix} k_1 \\ k_2 \\ k_3 \\ k_4 \\ k_5 \\ k_6 \\ k_7 \\ k_8 \end{pmatrix}
$$

where the macroscopic parameters on the left side are defined as below

$$\langle X \rangle = v_x t$$
$$\langle Y \rangle = v_y t$$
$$\langle (X - \langle X \rangle)^2 \rangle = 2D_x t$$
$$\langle (Y - \langle Y \rangle)^2 \rangle = 2D_y t$$
$$\langle (X - \langle X \rangle)(Y - \langle Y \rangle) \rangle = At$$
$$\langle (X - \langle X \rangle)^2 (Y - \langle Y \rangle) \rangle = Bt$$
$$\langle (X - \langle X \rangle)(Y - \langle Y \rangle)^2 \rangle = Ct$$
$$\langle (X - \langle X \rangle)^2 (Y - \langle Y \rangle)^2 \rangle - 2\langle (X - \langle X \rangle)(Y - \langle Y \rangle) \rangle^2$$
$$- \langle (X - \langle X \rangle)^2 \rangle \langle (Y - \langle Y \rangle)^2 \rangle = Et$$

With $k$-th displacements, $\Delta X_k = X_{k+1} - X_k$, $\Delta Y_k = Y_{k+1} - Y_k$,

$$v_x = \langle \Delta X \rangle / \Delta t$$
$$v_y = \langle \Delta Y \rangle / \Delta t$$

The covariance-based estimators for these (higher-order) covariances from the $x$- and $y$-displacement were calculated as below using the drift-adjusted coordinates

$Z = X - v_x t$, $W = Y - v_y t$,

$$2D_x = \left( \langle \Delta Z_k^2 \rangle + 2\langle \Delta Z_{k+1} \Delta Z_k \rangle \right) / \Delta t$$
$$2D_y = \left( \langle \Delta W_k^2 \rangle + 2\langle \Delta W_{k+1} \Delta W_k \rangle \right) / \Delta t$$
$$A = \left( \langle \Delta Z_k \Delta W_k \rangle + \langle \Delta Z_{k+1} \Delta W_k \rangle + \langle \Delta Z_k \Delta W_{k+1} \rangle \right) / \Delta t$$
$$B = \sum_{\substack{\alpha, \beta, \gamma = 0 \text{ or } 1 \\ \text{not } \alpha = \beta = \gamma = 1}} \langle \Delta Z_{k+\alpha} \cdot \Delta Z_{k+\beta} \cdot \Delta W_{k+\gamma} \rangle / \Delta t$$
$$C = \sum_{\substack{\alpha, \beta, \gamma = 0 \text{ or } 1 \\ \text{not } \alpha = \beta = \gamma = 1}} \langle \Delta Z_{k+\alpha} \cdot \Delta W_{k+\beta} \cdot \Delta W_{k+\gamma} \rangle / \Delta t$$
$$E = \sum_{\substack{\alpha, \beta, \gamma, \lambda = 0 \text{ or } 1 \\ \text{not } \alpha = \beta = \gamma = \lambda = 1}} f\left( \Delta Z_{k+\alpha}, \Delta Z_{k+\beta}, \Delta W_{k+\gamma}, \Delta W_{k+\lambda} \right) / \Delta t$$

where $f(x, y, z, w) = \langle xyzw \rangle - \langle xy \rangle \langle zw \rangle - \langle xz \rangle \langle yw \rangle - \langle xw \rangle \langle yz \rangle$.

Despite of the theoretically linear relationship between the macroscopic and microscopic parameters as above, simply solving the linear equation sometimes results in negative values of the rates, which should be 0 or positive. This is due to the uncertainties caused by the intrinsic stochastic nature of the random walk and the sensitivity of the higher-order covariances to the extreme values. Thus, we inferred the hopping rates and preferences as the posterior probabilities of the microscopic values that are limited to $k_i \geq 0$ and $0 \leq p_i \leq 1$ in a Bayesian model based on the assumption that the observed macroscopic parameters for each trajectory were obtained by random sampling from the Gaussian distributions around the theoretical values of the macroscopic parameters that are calculated from the microscopic parameters. Simulations with conditions similar to the current cases showed that this approach can distinguish between the different hopping patterns that exhibit the same $x$- and $y$- velocities and diffusion coefficients but distinct higher-order covariances. The Markov chain Monte Carlo sampling was implemented with Stan on R with uniform priors for all the parameters.

**3D-motility of a motor-coated bead on a surface-immobilized microtubule**. This was performed in flow chambers assembled from two coverslips using double-sided tape (NW-25, Nichiban) as spacers and essentially in the same way as detailed above. The flow chamber was sealed by nail polish. The longitudinal velocity of kinesins-coated bead movement on the microtubule was determined by fitting the $x$ displacement-time position of the bead with a linear function using Igor Pro 5.05 A.

**Microtubule gliding assay**. Microtubule gliding assays were performed in a flow chamber assembled from two coverslips using strips of double-sided tape (NW-25, Nichiban) as spacers. To activate the glass surface for immobilizing the His-tagged motor constructs, the chamber was first incubated with 5 mg ml$^{-1}$ recombinant Protein G (Thermo Fisher Scientific) in M buffer for 5 min at room temperature. The chamber was then washed with 4 chamber volumes of cold M buffer and 1 chamber volume of 50 ng μl$^{-1}$ anti-Penta·His antibody (QIAGEN) in M buffer was introduced. After 5 min incubation at room temperature, the chamber was washed with 4 chamber volumes of W buffer and a blocking buffer (1 mg ml$^{-1}$ casein in M buffer) was introduced. After 3 min incubation at room temperature, the chamber was washed with 4 chamber volumes of W buffer and 1 chamber volume of 1 μM solution of a motor construct in B buffer was introduced and incubated for 5 min at room temperature. Following a wash with 4 flow chamber volumes of W buffer, a chamber volume of a suspension of QDs (quantum dot 525 Streptavidin Conjugate, Thermo Fisher Scientific) -coated rhodamine-labeled microtubules[22] was introduced and incubated for 3 min at room temperature. Finally, after a wash with 2 chamber volumes of W buffer, 2 chamber volumes of the M buffer containing 0.5 mg ml$^{-1}$ casein, 2.7 mM ATP, 20 μM taxol, 0.5% β-mercaptoethanol, ATP regeneration and oxygen scavenger system were introduced. The flow chamber was sealed with nail polish, and the movements of microtubules at 23 °C were monitored by recording the 3D position of a QD every 0.4 s with the tPOT microscope. The 3D motion of QD was analyzed as described above for the movement of a motor-coated bead around a suspended microtubule. The pitch of the microtubule corkscrew motion was determined by fitting the $x$-$y$ position of the QD with a sine function using Igor Pro 5.05 A.

**Measurement of microtubule stimulated ATPase rate**. The microtubule-stimulated ATPase activities of individual kinesins were measured using a pyruvate kinase/lactate dehydrogenase-linked assay as described previously[59]. The activity of ATP hydrolysis coupled to NADH oxidation was monitored using 0.1–0.2 μM kinesins under the various microtubule concentrations ranging from 0.05 μM to 10 μM in M buffer containing 4.5 mM PEP, 2.7 mM ATP, 0.3 mM NADH, 2.5% PH-LDH (Sigma-Aldrich), 2.8% B buffer containing 80 mM NaCl and 18% BRB80t buffer (20 μM taxol, 80 mM PIPES, 1 mM MgCl$_2$, 1 mM EGTA, pH 6.8). The ATPase activities of individual kinesins were measured from three independent experiments under the same conditions. The turn-over rate and the Michaelis

constant were obtained by fitting the Michaelis–Menten equation to the kinetic data using Igor Pro 5.05 A.

**Statistics and reproducibility**. Statistical analysis of data was done using R. Kruskal–Wallis test with correction for multiple comparisons were used to compare the longitudinal velocities, rotational velocities and inverse of pitches, respectively, of M$_2$-, M$_2$C$_2$- and M$_1$-coated beads (Figs. 2, 3). Wilcoxon rank-sum test were used to compare the longitudinal velocities on the surface-immobilized microtubule and the suspended microtubule (Fig. 6). Tracking data for beads along suspended microtubules were obtained from at least 3 independent experiments on more than two separate days, and sample sizes and numbers are indicated in detail in the Results section.

**Reporting summary**. Further information on research design is available in the Nature Research Reporting Summary linked to this article.

## Data availability
All plasmids used in this study are available from the corresponding authors on request. The source data for graphs in the main figures are provided as Supplementary Data 1.

## Code availability
Custom scripts were written for determining protein concentration with ImageJ, for tracking kinesin-coated bead movements with Igor Pro 5.05 A, and for analysis of trajectories with R 3.5.3 and rstan 2.19.3. These scripts are available on reasonable request.

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

## Acknowledgements
We thank H. Takagi for advice on data analysis, and H. Yoshida for technical assistance with lithography, and I. Mishima for making grammatical corrections. This work was supported by JSPS KAKENHI (JP19H03190, JP20K06635 to J.Y.) by MEXT KAKENHI Grant-in-Aid for Scientific Research on Innovative Areas (JP19H05357, JP19H05378 to J.Y.) and by Cancer Research UK program grant (C19769/A11985 to M.M.).

## Author contributions
J.Y. and M.M. designed the project. Y.M., T.D., M.Y. and T.K. purified the proteins. T.O. and S.T. patterned the glass surface. M.S., Y.M., and S.Y. set up the microscopy work. Y.M. performed motility and ATPase assays. Y.M., M.Y., J.Y. and M.M. analyzed the motility data. All authors contributed to the writing of the manuscript.

## Competing interests
The authors declare no competing interests.
