## [Peer Review File · Communications Biology]

Reviewers' comments:

Reviewer #1 (Remarks to the Author):

This is a technical piece of work by Maruyama et al. that combines a clever experimental setup in which a single microtubule (MT) is suspended between two (2 micron) posts with the 3D tracking tPOT microscopy technique to study the centralspindlin complex and its motor component MKLP1. The authors employ nm resolution tracking of beads coated with MKLP1 monomer, dimer and centralspindlin (MKLP1 dimer plus a fragment of its binding partner CYK4) to measure their motility properties in 3 dimensions around the suspended MT. The experimental data is also compared to a mathematical model of motor stepping. Taken together, the authors conclude that dimeric MKLP1 intrinsically steps leftward at some frequency on the MT lattice and that this property is suppressed by the binding of CYK4, which renders it more comparable to monomeric MKLP1 that can stochastically side-step left and right along the lattice. While the experiments are well-executed and analyzed, the physiological relevance of these data are not entirely clear although the authors propose that the flexibility in off axis stepping conferred by CYK4 to MKLP1 would allow the centralspindlin complex to avoid obstacles along MTs in order to become enriched on the crowded MT environments in which centralspindlin is found in the cell.

I feel that the manuscript clearly meets three of the stated criteria for publication in Communications Biology namely: 1) novelty, 2) evidence supports the conclusions, 3) data are technically sound. The 4th criterion is that it is important to scientists in the sub-field of biology. In terms of the sub-field of motor biophysics this work is certainly important. However, I feel that further and proper contextualization and clarification of the physiological significance of these findings, and potentially a few additional experiments that can be done with existing reagents, are necessary to make this more relevant to the cytokinesis field.

Major concerns:

1) My major concern is that all of this work is done using beads that are coated with purified motor proteins/minimal centralspindlin complex. Therefore, the acquired data is reporting on the properties of motor ensembles on what is essentially a cargo (the bead). Yet, the the data is presented as if this is also relevant behavior of individual motor complexes on a single MT. For example, the schematic in Figure S1C shows an individual motor on the bead when the bead is actually fully coated with motors. This matters because 1) multiple bead-bound motors are likely associated with the MT at any given moment and 2) when a motor-detaches then a neighboring motor is very likely to associate with the MT in short order. Another example of this point is in the schematic in Figure 6F that shows individual dimers/centralspindlin complexes walking along the length of the MT. Finally, the physiological relevance envisioned by the authors also evokes the concept that individual centralspindlin complexes make long forays on the MT encountering many obstacles along their path that they must step around to make it to their ultimate destination.

2) In fact, how processive are individual centralspindlin complexes and MKLP1 dimers in this assay? The pitch data (0.8 μm and 1.2 μm) indicates that the motor is not very processive since a processive motor like kinesin-1 exhibits a significantly higher pitch (7 μm) in a similar assay. What is the run length of M2 and M2C2 in this assay? Given that there is a SNAP tag on these constructs it would be straightforward to label the purified protein with bright dyes such as the new Janelia Fluor dyes. The properties of brightly labeled M2 and M2C2 could then be measured in your setup and microscopy technique to measure the behavior of dimers rather than bead ensembles.

3) If the centralspindlin complex is not processive and exhibits short run lengths then I find the idea

that it needs to side-step obstacles along its path on a single MT to be much less appealing or physiologically relevant. Rather, the ensemble properties measured here are likely more relevant to centralspindlin's function/behaviors inside bundled MT arrays as alluded to briefly in the discussion with regards to the chirality of the central spindle and midbody. If one were to further flesh this bundle chirality idea out, how would this behavior affect the organization of parallel versus antiparallel bundles? Ultimately, I view centralspindlin's role in organizing bundled MT arrays as distinct and most likely more relevant than the proposed idea (and present physiological focus) that the complex walks over significant distances and side-steps obstacles along the way.

Other points:

1) It was unclear how the 1:1 mixture of 13mer (straight protofilaments) and 14mer (left-handed 6-8 um long helical pitch) is taken into account for correcting the pitch data that is presented. 50% of the data must come from beads associated with 13mers and 50% from 14mers. I presume that if one were to plot a histogram (like Figure 2C) for kinesin-1 coated beads you would see a bimodal distribution of events with a peak at $\sim 7\mu\text{m}$ and a peak at 0? Is a control baseline like this applied to the analysis of non-processive motors and torque-generating motors like what you are presumably studying here to attain the correct pitch?

2) On a related note, the first two sentences of the section "CYK4-binding switches double headed..." should appear earlier in the manuscript to allow the reader to better understand what this pitch value can mean. Also, the " $> \sim 6\mu\text{m}$ " pitch mentioned in the first sentence refers to the helicity of a 14 protofilament MT – correct. If so then this needs to be clarified.

Reviewer #2 (Remarks to the Author):

Maruyama et al. study the rotational motion of the kinesin-6 MKLP1, which is an important player in anaphase and cytokinesis. By extending the cutting-edge experimental setups from their previous works on other motor proteins, the authors found that dimeric MKLP1 moves around the microtubule making left-handed rotations, and that these rotations were less persistent in the presence of CYK4. Such flexible movement resulting from CYK4 binding may be important for avoiding obstacles, a hypothesis that the authors test elegantly by making spatial restrictions for the motor movement. Overall, this study is technically challenging, carefully executed, and led to important new knowledge about the motility and regulation of kinesin-6. I support publication of this work with minor changes.

The authors should briefly discuss the work by Ramaiya et al., PNAS 2017, about torque generation by kinesin-1.

It would be helpful to add a more thorough discussion of the mechanisms determining the handedness of motion, i.e., why MKLP1 (and canonical N-kinesins) follow a left-handed rather than a right-handed trajectory.

The authors write that CYK4 binding may induce a one-head-bound state. Here, the model from Mitra et al., PNAS 2018, should be discussed, where a one-head-bound state is related to sidestepping and helical motion. Do the authors think that a similar model may be relevant for centralspindlin? Does a one-head-bound state promote or reduce rotational movements? Does this depend on the type of kinesin?

Reviewer #3 (Remarks to the Author):

The manuscript by Maruyama and coauthors describes the influence of the co-factor CYK4 on the in vitro motility of microspheres coated with many kinesin-6 MKLP1 motor proteins. By 3D tracking, they found that binding of CYK4 to MKLP1 (forming a hetero-tetramer that resembles the centralspindlin complex found in vivo) reduced the angular diffusion constant and both the longitudinal and rotational speed of the motor-coated microspheres along freely suspended microtubules. Furthermore, in a 2D assay, they found more diffusive motion of hetero-tetramer coated microspheres. They interpret their multi-motor microsphere data using a single-molecule random-walk simulation and conclude that CYK4 promotes flexibility in off-axis forward stepping that may help to avoid obstacles on microtubules. While the data is interesting and technically sound, their interpretation is doubtful, and the simulation is rather speculative. My main concern is that microspheres are powered by many motors and the authors interpret their data as single motor experiments. Since there are many more possible interpretations for a multimotor scenario, the authors overinterpret their data and I strongly disagree with their conclusion as detailed below. Thus, I do not recommend publication of the manuscript in its present form. After a major revision, the manuscript may be considered.

Detailed comments:

1. The authors incubate their microspheres with micromolar motor concentrations. Thus, microspheres are fully decorated with motors. Therefore, during the measurements many motors interact simultaneously with the microtubule. Since motors are random steppers, they do not step in synchrony and exert both sideward and longitudinal forces onto each other. Also, motors attach and detach. Reattachment may be on a different protofilament resulting in microsphere rotations, which have nothing to do with sideward stepping. Rotational Brownian motion of the microsphere might add to the effect. Significant differences have been shown in similar assays between single and multimotor conditions. For example, for cytoplasmic dynein it was shown that single motors randomly stepped sideways while microspheres transported by many motors showed persistent unidirectional rotations (S. Can et al., eLife 3:e03205 (2014)). Similar effects have been observed for kinesin-8 (Ref. 23, 24 and 33). One possible explanation for the difference between single and multimotor conditions, is a bias in the force dependence of sideward steps (M. Bugiel et al., Biophys. J. 108, 2019 (2015)). Thus, using a multimotor assay to conclude on single-molecule parameters may lead to overinterpretation of the data. It needs to be clearly stated throughout the manuscript that multiple motors transport the microsphere. For instance in the abstract '...in a motility assay, we found that microspheres driven by many dimeric MKLP1 moved in a persistently left-handed...' The simulation of the motion with a single random stepper is meaningless and needs to be removed from the manuscript. All of the information is already contained in the MSD fits in Fig. S4. See further comments on this point below.
2. In the introduction, the authors should state the amino acid length of the neck region for MKLP1 and list the other N-kinesins that lack a conserved neck-linker sequence. Also, in the introduction a motivation for the 3D assays, the open question the authors want to address, and comparison to other approaches is missing. As an open question, the coordination between steps is mentioned, however, not addressed directly in the manuscript.
3. Page 5: the authors should state the bandwidth for the precision statements and add the temporal precision.
4. Page 5, line 14: Please provide the standard deviation of the temperature control.
5. The authors should add the affinity of CYK4 to MKLP1 is. For their M2C2 measurements, are all

motors actually in complex with CYK4? Since microspheres are washed, CYK4 is not directly attached to the microspheres, and CYK4 is in equilibrium with the motility buffer, some of the CYK4 molecules dissociate.

6. Page 7, line 3: The authors state to have worked with a 1:1 mixture of 13 and 14 protofilament microtubules. Since Ref. 19 contains many different protocols for microtubule preparation, the authors should be more specific on how they made their microtubules. Please provide a more detailed protocol in the methods. Some control measurements with kinesin-1 motors would be helpful to confirm the mixture.

7. Page 7, line 13: The authors compare the handedness of the helical trajectories of MKLP1 with that of other kinesins and conclude that all of them turn leftward. However, as pointed out in Point 1 above, for kinesin-8, it is unclear whether sideward steps are random or have a bias.

8. Are there single-molecule motility measurements with M2 and M2C2? Are speed and run length known for single molecules?

9. Fig. 2: Please change, 'Rotational revolutions' to 'Rotations' (also in Fig. 3). While the authors state the number of tested microspheres for the data shown, it is unclear, how many microspheres were tested in total. Did all microspheres that showed motility also show rotations? Did the authors discard traces? And if so, for what reasons?

10. Fig. 3D: There is a typo in 'Rotational velocity'.

11. Page 12: Power-law fitting of MSDs with non-integer exponents: Exponents $1 < \alpha < 2$ are attributed to anomalous diffusion which is not observed here. Power laws have a linear slope in a log-log plot, which is clearly not the case in Fig. 4B (apart fits and fit ranges are not shown or mentioned). Thus, there is no power law and anomalous diffusion here. It is a superposition of directed and random motion caused by the many motors attached to the microspheres. Such motion is well described by a parabola, which the authors used in Fig. S4. These fits should be shown in Fig. 4 on a log-log scale. Another reason for a non-linear slope of the log-log MSD plots, might be the presence of an offset expected from the tracking precision and data acquisition (see e.g. X. Michalet, Phys. Rev. E 82, 041914 (2010)). The authors should include an offset in their MSD fits. Please, also provide error bars on the fit parameters. An additional possibility for the nonlinear slopes, might be different populations of motors on the microspheres. For example, there might be a mixture of M2C2 with M2 that due to a low affinity of C2 are not present as a complex. Also, there might be different populations within M2C2. The authors could test whether the MSD values for a particular lag time are indeed normally distributed. Please show some histograms.

The random walk model has no justification and needs to be removed from the manuscript. It is essentially also a parabola and the fit parameters from Fig. S4 can simply be mapped onto the parameters of Eq. 1 & 2. There is no justification for the individual parameters of the model, e.g. the limitation of sideward steps to be diagonal, in particular since many motors interact simultaneously. Also the exponent of 2 for the longitudinal motion is no justification. The longitudinal motion is hardly affected by pure sideward steps. A pure sideward step simply reduces the average forward speed, but does not affect the longitudinal exponent. It would be reasonable to exclude backward steps.

Based on the MSD fit parameters of Fig. S4, one can calculate the average stepping rates due to sideward diffusion, directed sideward motion, total number of sideward steps, forward steps per second, and the ratio of forward to total number of sideward steps as listed in the respective columns below for the different conditions (all values are in steps per second assuming 8 nm for a forward and

28 deg for a sideward step):

M2 13 4.6 18 35 2.0

M2C2 5.4 1.6 7 20 2.8

M1 12 1.3 13 28 2.1

Based on this table, the total number of sideward steps per second is less for the M2C2 complex compared to M2 (7 versus 18). Thus, the conclusion that CYK4 promotes the flexibility in off-axis stepping is not supported by the data. On the contrary, sideward stepping is reduced! The different behavior in the 2D gliding assays (Fig. 6), could also be explained by a difference in the processivity. Ref. 3 suggests that CYK4 changes the motor conformation to promote microtubule bundling preventing processive motion of the motor on a single microtubule. Thus, if M2 is more processive than M2C2, it would rotate microspheres to the left until the microsphere hits the surface and then follow along the microtubule axis without further apparent rotational diffusion. Since the rotational bias towards the left is larger for M2, rightward steps in an ensemble of motors may not be resolved. For M2C2, only single heads are able to interact with the microtubule limiting processivity. Thus, there is more detachment/re-attachment kinetics that may cause the rotational diffusion. Therefore, the main conclusion of the manuscript is not supported by the data and the manuscript needs to be revised accordingly including the title. CYK4 binding seems to be important for the bundling of microtubules, but does not increase sideward stepping. Still obstacles may be avoided through detachment/re-attachment kinetics.

12. Fig. 5D, right: The '60' on the time axis is misaligned.

13. Fig. 6F: For consistency, also add the M2 and M2C2 nomenclature to the subplot.

14. Page 20, Table 1: Please provide raw data and standard errors for ATPase activity parameters K_{cat} and K_M . Please provide the total number of tested microspheres and list how many moved less than 1 micron, how many moved more than 1 micron and did/did not rotate. In particular the latter number is important. Please show exemplary traces of microspheres that did not rotate and include them in the data analysis of Fig. 2-4.

15. The pitch determination is unclear. Did the authors divide the total distance by the total number of rotations of an individual trace?

16. The discussion needs to be revised.

17. Page 31: Please provide more details on the quantum dots.

18. Page 32: Please state how microspheres were washed. The different motor concentrations for incubation suggest that different surface densities may have been present during experiments. Was the observed motility independent of the incubation concentrations?

19. Fig. S1: Please, also provide the precision for the radial position. How does the experimentally measured radius compare to the 140 nm estimate? Also, please, provide a reference or short comment about the radial and angular variance equations.

20. Fig. S2B: The tracks appear to be rather elliptical suggesting that the 3D tracking calibration was not very accurate.

Our response to the reviewers' comments

Original Reviewers' comments

Our response

Referee #1 (Remarks to the Author):

This is a technical piece of work by Maruyama et al. that combines a clever experimental setup in which a single microtubule (MT) is suspended between two (2 micron) posts with the 3D tracking tPOT microscopy technique to study the centralspindlin complex and its motor component MKLP1. The authors employ nm resolution tracking of beads coated with MKLP1 monomer, dimer and centralspindlin (MKLP1 dimer plus a fragment of its binding partner CYK4) to measure their motility properties in 3 dimensions around the suspended MT. The experimental data is also compared to a mathematical model of motor stepping. Taken together, the authors conclude that dimeric MKLP1 intrinsically steps leftward at some frequency on the MT lattice and that this property is suppressed by the binding of CYK4, which renders it more comparable to monomeric MKLP1 that can stochastically side-step left and right along the lattice. While the experiments are well-executed and analyzed, the physiological relevance of these data are not entirely clear although the authors propose that the flexibility in off axis stepping conferred by CYK4 to MKLP1 would allow the centralspindlin complex to avoid obstacles along MTs in order to become enriched on the crowded MT environments in which centralspindlin is found in the cell.

I feel that the manuscript clearly meets three of the stated criteria for publication in Communications Biology namely: 1) novelty, 2) evidence supports the conclusions, 3) data are technically sound. The 4th criterion is that it is important to scientists in the sub-field of biology. In terms of the sub-field of motor biophysics this work is certainly important. However, I feel that further and proper contextualization and clarification of the physiological significance of these findings, and potentially a few additional experiments that can be done with existing reagents, are necessary to make this more relevant to the cytokinesis field.

We appreciate this reviewer's supportive comments and constructive criticisms. Especially, we are very happy to hear that our manuscript meets all the criteria for publication in Communications Biology.

Major concerns:

1) My major concern is that all of this work is done using beads that are coated with purified motor proteins/minimal centralspindlin complex. Therefore, the acquired data is reporting on the properties of motor ensembles on what is essentially a cargo (the bead). Yet, the the data is presented as if this is also relevant behavior of individual motor complexes on a single MT. For example, the schematic in Figure SIC shows an individual motor on the bead when the bead is

actually fully coated with motors. This matters because 1) multiple bead-bound motors are likely associated with the MT at any given moment and 2) when a motor-detaches then a neighboring motor is very likely to associate with the MT in short order. Another example of this point is in the schematic in Figure 6F that shows individual dimers/centralspindlin complexes walking along the length of the MT. Finally, the physiological relevance envisioned by the authors also evokes the concept that individual centralspindlin complexes make long forays on the MT encountering many obstacles along their path that they must step around to make it to their ultimate destination.

We understand this reviewer's concern that our ways of presenting our findings might have been misleading. This could be partly because we failed to emphasize the physiological importance of the higher-order assembly of centralspindlin heterotetramers into oligomers or clusters and because our schematics were not precise.

Working as a self-organized motor ensemble is an intrinsic property of centralspindlin. As we previously reported (Hutterer et al. 2009), centralspindlin travels along the interpolar microtubules as oligomers/clusters of motors during anaphase. Thus, the comments by this reviewer,

This matters because 1) multiple bead-bound motors are likely associated with the MT at any given moment and 2) when a motor-detaches then a neighboring motor is very likely to associate with the MT in short order.

exactly illustrate the way how we believe centralspindlin moves along microtubules towards the plus-ends before it reaches the equatorial zone of anti-parallel microtubule overlaps. Although we believe that the difference in the motility of the beads coated with multiple different MKLP1 constructs (eg. M₂ vs M₂C₂) somehow reflects their difference at the individual molecule levels (in this case, the presence vs absence of CYK4), we agree that the effect might be indirect due to the added layer of complexity by the involvement of the multiple motors and thus we should be more careful about interpreting our data.

In this revised version, we a) rewrote the introduction to clarify the physiological relevance of the multi-motor ensembles of centralspindlin, b) described our experimental conditions more carefully to make the involvement of multiple motors in the bead motion more explicit, c) entirely rewrote the Discussion section, d) revised the legend for Figure S1C, and e) removed Figure 6F.

a. The latter half of the second paragraph of Introduction (page 3 lines 51 – 60) now reads

For the functions of centralspindlin in cytokinesis, its timely and sharp accumulation to the center of the central spindle and the midbody is essential (Matulienė and Kuriyama 2002; Minestrini, et al. 2003; Hutterer et al. 2009; Douglas et al. 2010). This relies on its plus-end directed motility, which is modulated by the CYK-binding to the neck of MKLP1 and by further assembly of the heterotetramers into multimeric clusters. CYK4-binding to the neck reconfigures the motor heads

in the heterotetrameric centralspindlin suitable for anti-parallel bundling (Davies et al. 2015). It also slows down the motility in the surface gliding assay (White et al. 2013; Davies et al. 2015). Higher-order clustering promotes the plus-end directed transport without detachment from the microtubule (Hutterer et al. 2009).

b. We clarify the multiple motor conditions of the MKLP1-coated bead (page 5 lines 98 – 101)

In a preliminary screening for a condition that allows us to observe stable motion of the bead along the microtubule, we realized that the bead needed to be coated with multiple motors. We hereafter used the minimum motor:bead ratio that allowed a stable motion although we don't know exactly how many motors are simultaneously interacting with a microtubule at any given time.

c. In Discussion, when we describe our speculations about the possible molecular mechanisms for the modulation of directional preferences, we tried to be cautious about the complexity due to the multiple motor condition. For example, page 25 lines 432 – 435 now read

CYK4-binding might interfere with this head-to-head communication in the dimer and thus relaxes the bias in the protofilament switching although we should be cautious upon linking our observation to a structural mechanism since multiple motors are involved in the helical motion of a bead in our experiments.

2) In fact, how processive are individual centralspindlin complexes and MKLP1 dimers in this assay? The pitch data (0.8 μm and 1.2 μm) indicates that the motor is not very processive since a processive motor like kinesin-1 exhibits a significantly higher pitch (7 μm) in a similar assay. What is the run length of M2 and M2C2 in this assay? Given that there is a SNAP tag on these constructs it would be straightforward to label the purified protein with bright dyes such as the new Janelia Fluor dyes. The properties of brightly labeled M2 and M2C2 could then be measured in your setup and microscopy technique to measure the behavior of dimers rather than bead ensembles.

We highly appreciate the reviewers' insightful suggestion to use the novel dye that can be introduced into the existing constructs via the SNAP tag. However, our current system (*tPOT*) for the 3D nanometry is not compatible with the TIRF illumination of a laser beam and thus not powerful enough to detect a single fluorophore. Thus, unfortunately, it is impossible to determine the single molecule behavior in our assay as suggested.

The mean run-length of non-clustered MKLP1 dimer (clustering defective Z585GFP, similar to but slightly longer than M₂) measured on the surface-immobilized microtubules is 150 nm while it is increased to ≥ 800 nm by higher-order assembly into clusters (clustering-competent Z601GFP or Z585GFP artificially clustered with avidin) (Hutterer et al. 2009). M₁ is a single-headed construct and thus it is highly likely that it is not a processive motor. We observed the

processive motility of both the clustered full-length centralspindlin complex in vivo and the clusters of Z601GFP+C₂ (similar to the current M₂C₂) in vitro (Davies et al. 2015). However, it is reasonable to assume that, without clustering, M₂C₂ would be a non-processive motor since CYK4-binding to the neck of MKLP1 likely prevents the two heads in the heterotetramer from simultaneously binding to the same microtubule.

We agree with this reviewer on the importance of measurement of the processivity or the run length as basic characteristics of a molecular motor. However, considering the physiological significance of the clustering, **it would make sense only when we could determine these parameters as functions of the degree of clustering**. This would need a lot of effort and is beyond the scope of our current work.

As for the relationship between processivity and helical pitch, Brunnbauer et al (2012) showed that the kinesin-driven helical pitch was not directly linked to its processivity. For example, kinesin-2 motors CeKLP11/20 and kinesin-8 motor kip3 (Mitra et al. 2018) are highly processive motors and their helical pitches are ~0.6 μm and ~2 μm , respectively, which were less than 6-8 μm of pitch driven by kinesin-1.

3) If the centralspindlin complex is not processive and exhibits short run lengths then I find the idea that it needs to side-step obstacles along its path on a single MT to be much less appealing or physiologically relevant. Rather, the ensemble properties measured here are likely more relevant to centralspindlin's function/behaviors inside bundled MT arrays as alluded to briefly in the discussion with regards to the chirality of the central spindle and midbody. If one were to further flesh this bundle chirality idea out, how would this behavior affect the organization of parallel versus antiparallel bundles? Ultimately, I view centralspindlin's role in organizing bundled MT arrays as distinct and most likely more relevant than the proposed idea (and present physiological focus) that the complex walks over significant distances and side-steps obstacles along the way.

As a signaling hub for cytokinesis, accumulation of centralspindlin to a narrow zone of cell equator is crucial for cytokinesis. We can observe largely unidirectional motion of the clusters of centralspindlin in dividing mammalian cultured cells, which can be as long as 2 μm (Hutterer et al. 2009). Motor-dead MKLP1 with a mutation in the ATP-binding site fails to tightly accumulate to the spindle midzone (Matulienė et al. 2002; Minestrini et al. 2003). Accumulation of wild-type centralspindlin to the tips of equatorial astral microtubules has also been reported (Nishimura & Yonemura 2006). These clearly indicate the physiological importance of the efficient transport of centralspindlin as a self-organized team of motors in the highly crowded environment of the mitotic spindle. To clarify this point, we now rewrote the introduction as mentioned in our response to point 1).

As correctly pointed out by this reviewer, anti-parallel bundling of microtubules is another crucial function of centralspindlin. Although we discussed an implication of our finding for avoiding unnecessary twist of the central spindling and midbody in the original version, we didn't consider the implications of the torque generation by a crosslinking motor for the parallel vs anti-parallel configuration of microtubule bundles. We thank this reviewer for pointing this out. We now discuss this possibility in Discussion (page 27 lines 466 – 477).

An alternative, but not exclusive, possibility would be the relaxation of the torque applied to the spindle. A torque-generating and crosslinking minus-end directed kinesin, kinesin-14, slides parallel and anti-parallel microtubule overlaps in different ways and induces a twist in a locked anti-parallel bundle (Fink et al. 2009; Mitra 2020). The torque generation, in addition to the optimized head configuration by CYK4 (Davies et al. 2015), might contribute to preferred accumulation of centralspindlin to the anti-parallel overlaps pre-templated before anaphase onset. Torque by kinesin-5, a plus-end directed motor and crosslinker, introduces a helical twist onto the microtubule organization of the mitotic spindle (Novak 2018). CYK4-binding, which relaxes the directional preferences of the MKLP1 dimer, might contribute to avoiding the helical twists of the central spindle and the midbody. Future studies with higher spatial and temporal observations of the in vivo dynamics of centralspindlin and the microtubules would be necessary to reveal the biological roles of stochastic protofilament switching.

Other points:

1) It was unclear how the 1:1 mixture of 13mer (straight protofilaments) and 14mer (left-handed 6-8 um long helical pitch) is taken into account for correcting the pitch data that is presented. 50% of the data must come from beads associated with 13mers and 50% from 14mers. I presume that if one were to plot a histogram (like Figure 2C) for kinesin-1 coated beads you would see a bimodal distribution of events with a peak at ~7um and a peak at 0? Is a control baseline like this applied to the analysis of non-processive motors and torque-generating motors like what you are presumably studying here to attain the correct pitch?

There is a known equation between the pitch of observed helical motion (P_{observed}), the pitch of the helical arrangement of protofilaments around the microtubule axis ($P_{\text{MT}} = \infty$ ($n = 13$) or $6 \sim 8 \mu\text{m}$ ($n = 14$)), and the corrected pitch of the helical motion of a motor (P_{motor}) (Brunnbauer et al. 2012; Can et al. 2014; Mitra et al. 2015)

$$\frac{1}{P_{\text{obs}}} = \frac{1}{P_{\text{motor}}} + \frac{1}{P_{\text{MT}}}$$

Using this, we can calculate the mean observed pitch (P_{obs}) of the trajectories of a motor, whose intrinsic pitch as to the protofilaments is P_{motor} , on the $\alpha:(1-\alpha)$ mixture of 13- and 14- protofilament microtubules

$$\begin{aligned}
P_{\text{obs}} &= \alpha P'_{\text{obs}} + (1 - \alpha) P''_{\text{obs}} \\
P'_{\text{obs}} &= P_{\text{motor}} \\
P''_{\text{obs}} &= \frac{1}{1/P_{\text{motor}} + 1/P_{\text{MT}}}
\end{aligned}$$

where P'_{obs} and P''_{obs} are the pitches that would be observed if all the microtubules uniformly had 13- and 14- protofilaments, respectively. By solving this, we get

$$P_{\text{motor}} = \frac{P_{\text{obs}} - P_{\text{MT}} + \sqrt{(P_{\text{obs}} - P_{\text{MT}})^2 + 4\alpha P_{\text{obs}} P_{\text{MT}}}}{2\alpha}$$

With $\alpha = 5/8$, $P_{\text{MT}} = 8 \mu\text{m}$, we get

	$P_{\text{obs}}/\mu\text{m}$	$P_{\text{motor}}(\alpha = 5/8)/\mu\text{m}$	difference
M2	0.8	0.829	3.6%
M2C2	1.5	1.600	6.7%
M1	2.1	2.291	9.1%

This indicates that the influence of the heterogeneity in the protofilament number on the correction of the helical pitch is relatively small, compared to the differences between the different MKLP1 constructs. We mention this in the Methods section (pages 32-33, lines 602 – 612).

We also considered the influence of the heterogeneity of the protofilament numbers on the inference of the directional preferences. The results of the calculation assuming $n = 13$ or 14 are presented in Figure 5 and Figure S4, respectively. They look pretty similar in that M2C2 is more similar to M1 than to M2 and that the most prominent differences between M2 vs M2C2 are found in front-right (p_3) and backward (p_7).

2) On a related note, the first two sentences of the section “CYK4 -binding switches double headed...” should appear earlier in the manuscript to allow the reader to better understand what this pitch value can mean. Also, the “> ~6 um” pitch mentioned in the first sentence refers to the helicity of a 14 protofilament MT – correct. If so then this needs to be clarified.

We have now mentioned it earlier in the manuscript and corrected the sentence refers to the helicity of microtubules as the reviewer suggests. (page 7 line 113 – page 8 line 128)

Our preparation of microtubules is a 5:3 mixture of microtubules made of 13-protofilaments and those of 14-protofilaments (Yajima et al. 2008). While protofilaments in a 13-protofilament microtubule are parallel to the axis of the microtubule, in a 14-protofilament microtubule, they are twisted in a left-handed helix around the microtubule axis with a long helical pitch (6~8 μm) (Ray et al. 1993). If an MKLP1-coated bead accurately tracked a single protofilament without

protofilament switching like double-headed kinesin-1 (Ray et al. 1993; Fehr et al. 2008; Mitra et al. 2015; Bugiel et al. 2018), it would have shown a helical trajectory with a pitch and handedness identical to the supertwist of the protofilaments around the microtubule axis, i.e., no helicity on a 13-protofilament microtubule or a left-handed helix with a much longer pitch on a 14-protofilament one. On the other hand, if every step was made onto the binding site at the left-front on the microtubule lattice, the helical pitch would be $\sim 0.1 \mu\text{m}$ ($\sim 13 \times 8 \text{ nm}$). The observed helical pitches of the MKLP1-driven bead motions were between these values (Figure 2B). This indicates that neither the supertwist of the protofilaments nor invariable left-front stepping alone can explain the observed helical motion driven by MKLP1. Rather it suggests that MKLP1 kinesin-6 exerts a mechanical force that consists of not only the axial component along a protofilament, but also the torque-generating, off-axis component, which causes occasional protofilament switching

Reviewer #2 (Remarks to the Author):

Maruyama et al. study the rotational motion of the kinesin-6 MKLP1, which is an important player in anaphase and cytokinesis. By extending the cutting-edge experimental setups from their previous works on other motor proteins, the authors found that dimeric MKLP1 moves around the microtubule making left-handed rotations, and that these rotations were less persistent in the presence of CYK4. Such flexible movement resulting from CYK4 binding may be important for avoiding obstacles, a hypothesis that the authors test elegantly by making spatial restrictions for the motor movement. Overall, this study is technically challenging, carefully executed, and led to important new knowledge about the motility and regulation of kinesin-6. I support publication of this work with minor changes.

We appreciate this reviewer for his/her high evaluation of our work.

The authors should briefly discuss the work by Ramaiya et al., PNAS 2017, about torque generation by kinesin-1.

We thank this reviewer for pointing out this. The torque by dimeric kinesin-1 discussed in Ramaiya et al. (2017) is the one around its stalk that causes the rotation/spin of the attached microsphere. A special setup is required to detect this type of rotation or spin. This is distinct from the torque that has been reported for kinesins-2, 5, 8 and we studied in this manuscript, which is the force component perpendicular to the microtubule axis and causes rotation/revolution of the attached marker around the microtubule axis. This can be detected by tracing the 3D position of the marker. We mention this difference when we first mention the torque generation by MKLP1 in Results, page 8 lines 126 – 130.

Rather it suggests that MKLP1 kinesin-6 exerts a mechanical force that consists of not only the axial component along a protofilament, but also the torque-generating, off-axis component, which causes occasional protofilament switching. Note that this torque is distinct from the one that causes the rotation/spin of the marker attached to the tail of dimeric kinesin-1 (Ramaiya et al. 2017).

It would be helpful to add a more thorough discussion of the mechanisms determining the handedness of motion, i.e., why MKLP1 (and canonical N-kinesins) follow a left-handed rather than a right-handed trajectory.

All the N-kinesins tested so far, including kinesins-1, generate a left-handed torque around the microtubule axis in their monomeric form. Dimerization seems to modulate this (eg. no torque of this kind by dimeric kinesin-1). Unfortunately, the mechanisms for the handedness remain unclear. In the revised Discussion, we list various models proposed so far (see below).

The authors write that CYK4 binding may induce a one-head-bound state. Here, the model from

Mitra et al., PNAS 2018, should be discussed, where a one-head-bound state is related to sidestepping and helical motion. Do the authors think that a similar model may be relevant for centralspindlin? Does a one-head-bound state promote or reduce rotational movements? Does this depend on the type of kinesin?

To address this and the above points, we now discuss the possible mechanisms for the torque generation by monomeric and dimeric kinesins separately in analogy with the proposed mechanisms for the better studied axial force generation along the microtubule axis. In this part, we mentioned Mitra et al. (2018) as a variant of the diffusive search of the leading head mechanism. (page 25 lines 419 – 424)

..., the mechanisms proposed for the head-to-head communication in the coordinated motion of dimeric kinesin-1 along the microtubule axis might explain the modulation of the torque as well. These include the diffusive search of the leading head (Yajima and Cross 2005; Bormuth et al. 2012; Mitra et al. 2018; 2019) and anisotropic detachment of the trailing head via intramolecular strain (Uemura et al. 2002; Bugiel et al. 2015; McHugh et al. 2018).

We then discussed our speculations on the torque generation by MKLP1. (page 25 lines 426 – 435)

We have found that centralspindlin (heterotetramer of MKLP1 and CYK4) behaves more like monomeric MKLP1 than dimeric MKLP1 in the microtubule-activated ATPase and in the directional preference of the stochastic helical motion around a microtubule. Our previous work revealed that CYK4 binding to the neck of MKLP1 makes the configuration of the two motor domains in a dimer less flexible and thus prevents their simultaneous binding to the same microtubule (Davies et al. 2015). We speculate that the presence of another motor head flexibly linked via long necks might facilitate the leftward torque generated by a motor head to be translated into the proto-filament switching towards the left. CYK4-binding might interfere with this head-to-head communication in the dimer and thus relaxes the bias in the protofilament switching although we should be cautious upon linking our observation to a structural mechanism since multiple motors are involved in the helical motion of a bead in our experiments.

Reviewer #3 (Remarks to the Author):

The manuscript by Maruyama and coauthors describes the influence of the co-factor CYK4 on the in vitro motility of microspheres coated with many kinesin-6 MKLP1 motor proteins. By 3D tracking, they found that binding of CYK4 to MKLP1 (forming a hetero-tetramer that resembles the centralspindlin complex found in vivo) reduced the angular diffusion constant and both the longitudinal and rotational speed of the motor-coated microspheres along freely suspended microtubules. Furthermore, in a 2D assay, they found more diffusive motion of hetero-tetramer coated microspheres. They interpret their multi-motor microsphere data using a single-molecule random-walk simulation and conclude that CYK4 promotes flexibility in off-axis forward stepping that may help to avoid obstacles on microtubules. While the data is interesting and technically sound, their interpretation is doubtful, and the simulation is rather speculative.

My main concern is that microspheres are powered by many motors and the authors interpret their data as single motor experiments. Since there are many more possible interpretations for a multimotor scenario, the authors overinterpret their data and I strongly disagree with their conclusion as detailed below. Thus, I do not recommend publication of the manuscript in its present form. After a major revision, the manuscript may be considered.

We took this reviewer's criticisms seriously and made significant revisions especially on the mathematical modeling and the interpretation of the data detailed as below.

Detailed comments:

1. The authors incubate their microspheres with micromolar motor concentrations. Thus, microspheres are fully decorated with motors. Therefore, during the measurements many motors interact simultaneously with the microtubule. Since motors are random steppers, they do not step in synchrony and exert both sideward and longitudinal forces onto each other. Also, motors attach and detach. Reattachment may be on a different protofilament resulting in microsphere rotations, which have nothing to do with sideward stepping. Rotational Brownian motion of the microsphere might add to the effect. Significant differences have been shown in similar assays between single and multimotor conditions.

We understand the thick accumulation of work on the mechanisms for the motility by kinesins (especially, kinesin-1) and dynein by observing and manipulating them under the single-motor condition. We also fully acknowledge that our assays were done under the condition that a bead is coated with multiple motors. We agree with this reviewer that in the multiple-motor condition, there are more factors to be considered than in the single-motor condition, although we failed to explicitly discuss this in our original manuscript. As listed up by this reviewer, these include asynchronous but potentially simultaneous interaction with the microtubule, mechanical communication via the microtubule, rebinding with different motors associated with rotation of the bead, etc. To address this point, in the current manuscript, we tried to make it clear throughout the manuscript that our measurements were done under the multiple motor

conditions. We discussed the complexities introduced by this setup and the limitations in our interpretation. Please also refer to our response to the major point 1 raised by Reviewer #1.

Here we wish to emphasize two points important for the relevance of our work. The first one is that the plus-end-directed transport as a self-organized multiple motor ensemble is an intrinsic property of centralspindlin that is crucial for cytokinesis. We can observe largely unidirectional motions of the clusters of centralspindlin in dividing mammalian cultured cells, which can be as long as 2 μm (Hutterer et al. 2009). A mutant ZEN-4 (*C. elegans* MKLP1) defective for clustering of centralspindlin fails to sharply accumulate to the division plane and causes abortion of cytokinesis. Clustering is also required for the processive motility of MKLP1 in an in vitro motility assay (Hutterer et al. 2009). Thus, studying the motility of MKLP1 as an ensemble of multiple motor molecules and the influence of the unique neck-binding subunit CYK4 is of relevance biologically and biophysically.

The second point is that the beads coated with all the three MKLP1 constructs **exhibited left-handed helical trajectories**, which can't be explained by the possible twist of the protofilaments, **in the absence of external forces**. In other words, the MKLP1-coated beads **spontaneously broke the left-right symmetry** as well as forward-backward symmetry through the interaction between the motors and the microtubule surface. Importantly, the beads driven by different constructs showed different behaviors, implying the role of the difference between them in the symmetry breaking.

For example, for cytoplasmic dynein it was shown that single motors randomly stepped sideways while microspheres transported by many motors showed persistent unidirectional rotations (S. Can et al., eLife 3:e03205 (2014)).

We compared MKLP1 constructs using a bead of the same size (220 nm diameter) as a reporter. In contrast, in Can et al. (2014), observation of the 'multiple motor' condition was performed using a large microsphere of ~500 nm diameter while the 'single motor' condition was with a quantum dot (QD) of ~20 nm diameter. Since, according to Stokes' law, the drag coefficient of a spherical object is proportional to its radius, the drag coefficient of the QD is 25-times smaller than that of the bead. According to the Einstein-Smoluchowski relation, this means that, at the same temperature, the diffusion coefficient is 25-times bigger for the QD than the bead. Anchorage to a microtubule via dynein would alleviate this thermal fluctuation. However, this stabilizing effect depends on the binding of dynein to the microtubule, and the structural rigidities of dynein itself, and the dynein-HaloTag and HaloTag-QD joints, and thus isn't likely to be stronger for the QD than for the larger microsphere. The bigger thermal agitation on the motion of the QD would result in more frequent switching of the direction of the motion and might have obscured the bias in the torque generated by the single-molecule of dynein. Thus, we can't agree with this reviewer on treating Can et al. (2014) as an established, strong piece

of evidence for the emergence of the persistent torque by an ensemble effect of motors that neither exert a torque nor show a bias in protofilament switching as a single motor.

Similar effects have been observed for kinesin-8 (Ref. 23, 24 and 33).

Borumuth et al. (2012) (ref. 23 in the original manuscript) reported the torque generation by Kip3 kinesin-8 in the surface gliding assay, i.e., under the multiple motor condition. Mitra et al. (2018) (ref. 24 in the original manuscript) not just confirmed this but also observed that the single-molecule of Kip3 labeled with a QD showed a helical motion (and thus the side-stepping) around a suspended microtubule (note that Kip3 is a highly processive motor and thus its interaction with a microtubule is much more stable than dynein). In contrast, Bugiel and Schäffer (2018) (ref. 33 in the original manuscript) failed to detect the persistent helical motion around a suspended microtubule irrespective of the numbers of the motor per bead. Although the exact reason for this apparent discrepancy remains unclear, a big difference is that Bugiel and Schäffer used a feedback-controlled optical trap to follow the motion of a bead coated with a single or multiple Kip3 molecules. As they mentioned in the Note Added in Proof, the small force inevitably applied by the trap might have interfered with the rotational motion.

In summary, there is no compelling evidence for the emergence of torque by an ensemble of the motors that do not generate torque at all as a single molecule although, of course, this does not exclude the possible complex effects due to the involvement of multiple motors. A moderate point of view would be that monomeric kinesins often generate a torque but this can be modulated by dimerization and involvement of multiple motors. We believe that a difference in the results obtained with different MKLP1 constructs is likely to be somehow reflecting the difference between them.

One possible explanation for the difference between single and multimotor conditions, is a bias in the force dependence of sideward steps (M. Bugiel et al., Biophys. J. 108, 2019 (2015)). Thus, using a multimotor assay to conclude on single-molecule parameters may lead to overinterpretation of the data.

The positions and orientations of the MKLP1 constructs attached on the bead surface should be random. If so, it is reasonable to assume that there is no sophisticated coordination between multiple MKLP1 molecules. In this circumstance, it is very difficult to explain the observed spontaneous leftward bias just by the multiple-motor effect without any bias that individual motors might somehow exert. By contrast, there are clear structural differences between the constructs we tested such as the confined head-to-head configuration introduced by CYK4-binding to the neck of MKLP1. Thus, we don't think it is a totally stupid idea to try to explain our observations by the possible difference in the behaviors of different MKLP1 constructs although, as this reviewer pointed out and we discussed above, we need to be cautious about the complexities due to the involvement of multiple motors.

In this version, we discussed the model proposed by Bugiel et al. (2015) as a possible mechanism for the bias in the protofilament switching and the torque generation.

It needs to be clearly stated throughout the manuscript that multiple motors transport the microsphere. For instance in the abstract '...in a motility assay, we found that microspheres driven by many dimeric MKLP1 moved in a persistently left-handed...'

We agree that we failed to properly explain what we aimed for in our previous mathematical modelling and that our way of describing our interpretations was ambiguous and misleading. We now changed the sentences as the reviewer suggested (page 2 lines 22 – 23). We also now mentioned the importance of clustering of centralspindlin for its biological functions (page 3 line 51~ page 4 line 60) and added/modified the sentences;

page 4 lines 72 – 74.

Here we investigate whether a team of MKLP1 kinesin-6 motors can switch protofilaments during its plus-end-directed motion by 3D nanometry of the motion of a bead coated with multiple motors on a freely suspended microtubule.

page 5 lines 82 – 83.

To gain insight into the protofilament switching during the plus-end-directed motion of centralspindlin, we examined the 3D motion of a bead driven by multiple MKLP1 kinesin-6

page 5 lines 98 – 99.

In a preliminary screening for a condition that allows us to observe stable motion of the bead along the microtubule, we realized that the bead needed to be coated with multiple motors.

page 11 lines 176 – 179.

...the beads coated with multiple M_1 showed predominantly left-handed helical trajectories with an average longitudinal velocity of $0.24 \pm 0.04 \mu\text{m s}^{-1}$, an average rotational velocity of $0.11 \pm 0.09 \text{ rev s}^{-1}$ and an average pitch of $2.1 \pm 1.6 \mu\text{m}$, respectively (mean \pm SD)

page 25 lines 434 – 435.

...although we should be cautious upon linking our observation to a structural mechanism since multiple motors are involved in the helical motion of a bead in our experiments.

page 6 Figure 1C legend.

Microtubules were suspended between them using streptavidin-biotin interaction. This allowed beads coated with multiple kinesin molecules to move all around the microtubule surface.

page 7 Figure 2A legend.

The beads coated with multiple M_2C_2 and M_1 occasionally reversed their rotational direction for a short period of time.

page 7 Figure 2B legend.

The beads coated with multiple M_2C_2 or M_1 more frequently exhibited longer pitches than the beads coated with M_2 due to occasional reversal of the rotational direction.

The simulation of the motion with a single random stepper is meaningless and needs to be removed from the manuscript.

We regret that we were not careful enough about distinguishing between the motion of the bead and the stepping of a motor to a nearby site. These two can be treated as identical in principle (putting aside the thermal agitation of the bead) under the single-molecule condition, but not in the multiple motor condition. We agree with this reviewer that using our data to model the stepping of a motor is inappropriate. As we discussed above, the involvement of multiple motors adds a layer of complexity. However, we believe that it is still reasonable to use the random walk framework to characterize the average behaviors such as the directional preference of the bead motion driven by multiple motors on the 2-dimensional lattice surface of the microtubule. In the revised manuscript, we clarified that the purpose of our model is to better describe the motion of the bead, not directly parametrize the stepping of the motor. For example, the “Modeling the stochastic helical motion as a biased random walk on the MT surface lattice” section (page 16 line 262) now starts with:

The **fluctuating helical motions of the beads** driven by the MKLP1 constructs can be viewed as a 2D random walk with anisotropic/biased hopping rates on the microtubule lattice surface.

All of the information is already contained in the MSD fits in Fig. S4. See further comments on this point below.

The longitudinal and rotational MSD plots in Fig. 4S of the original manuscript (and their updates considering the heterogeneity in the mode of motion in Fig. 4 H and I of the revised manuscript) illustrate the dependence to the time lag of a combination of the constant drift and uncertainties in the position of the bead due to diffusion and measurement **in one dimension**, i.e., either along the microtubule axis or towards the sideward on the microtubule surface. The statement “*All of the information is already contained in the MSD fits in Fig. S4*” is equivalent to making a very big assumption that the protofilament switching is totally independent of the motion along the protofilament axis, which has not been proven for any type of torque generating microtubule motor. Considering that the motions in two directions are originated from the mechanical action of the same motor, we can’t simply assume their independence. In this revision, we analyzed our data with a newly developed model that consider the potential correlations between the motions along the two axes as detailed in our response to this reviewer’s point 11.

2. In the introduction, the authors should state the amino acid length of the neck region for MKLP1 and list the other N-kinesins that lack a conserved neck-linker sequence.

In the revised manuscript, we now mention the amino acid length of the neck region of MKLP1 (page 3 line 40). We also mention kinesins-10 and -11 as additional examples of N-kinesins with an atypical neck structure (page 3 line 38). A more comprehensive listing is found in our previous publication (Davies T et al. 2015. Figure 8 and figure S1).

Also, in the introduction a motivation for the 3D assays, the open question the authors want to address, and comparison to other approaches is missing. As an open question, the coordination between steps is mentioned, however, not addressed directly in the manuscript.

We agree with this reviewer that our motivation was not very unclear. To place our work into more proper biological contexts, we rewrote Introduction and Discussion, trying to clarify that working as an ensemble of multiple motors is an intrinsic and biological crucial characteristic of centralspindlin and that the crowdedness of the mitotic spindle can be an obstacle against the motor-dependent equatorial accumulation of centralspindlin, which is crucial for cytokinesis. Please refer to our response to Reviewer #1's point 1.

3. Page 5: the authors should state the bandwidth for the precision statements and add the temporal precision.

For a typical 0.1 s cycle, images were acquired using an EM-CCD camera every 100.28 ms with 100 ms exposure. This information is added in Methods (page 30, lines 551 – 553). The fluctuation in the cycle time was negligible (SD < 100 ns, according to the time stamps).

4. Page 5, line 14: Please provide the standard deviation of the temperature control.

This is now described in Methods (page 31 line 559, “23.0 ± 0.2°C (mean ± SD)”).

5. The authors should add the affinity of CYK4 to MKLP1 is. For their M2C2 measurements, are all motors actually in complex with CYK4? Since microspheres are washed, CYK4 is not directly attached to the microspheres, and CYK4 is in equilibrium with the motility buffer, some of the CYK4 molecules dissociate.

The affinity between CYK4 and MKLP1 was measured previously with similar constructs (Pavicic-Kaltenbrunner et al. (2007)). The dissociation constant, K_D , is 4 nM at 16 °C and 10 nM at 25 °C. The rate of dissociation, k_{off} , is $3.4 \times 10^{-4} \text{ s}^{-1}$ at 16 °C and $8.7 \times 10^{-4} \text{ s}^{-1}$ at 25 °C. The rates of association, k_{on} , is $8.6 \times 10^4 \text{ mol}^{-1} \text{ s}^{-1}$ at 16 °C and $9.13 \times 10^4 \text{ mol}^{-1} \text{ s}^{-1}$ at 25 °C.

We can estimate the rates and equilibrium constant in our 3D nanometry condition at 23 °C using the Arrhenius equation although the buffer condition is slightly different. To prepare the M2C2-coated beads, 4.1 μM M2C2 was incubated with the beads and the sedimented beads were resuspended into a 25× volume of the assay buffer. Assuming that all the M2C2 molecules were trapped on the beads, after this wash/dilution step, the total concentration of M2C2 was 164 nM, 20 times higher than K_D . The kinetics of dissociation can be predicted by solving

$$\frac{d[M_2C_2]}{dt} = k_{on}[M_2][C_2] - k_{off}[M_2C_2]$$

The time course of the fraction of M₂C₂ maintained as the complex will look like the graph below

Thus, the majority of M₂C₂ (>80% at least) must have remained as the complex during our assay. This is consistent with the observed behavior of the M₂C₂ beads since they would have looked much more like the M₂ beads if the majority of M₂C₂ was dissociated apart. This is mentioned in the revised text (page 5 line 101 – page 6 line 104).

Considering the high affinity between MKLP1 and CYK4 ($K_D \sim 7$ nM) (Pavicic-Kaltenbrunner, et al. 2007), it is likely that the majority of MKLP1 on the M₂C₂-coated beads are kept associated with CYK4 in our experimental condition.

6. Page 7, line 3: The authors state to have worked with a 1:1 mixture of 13 and 14 protofilament microtubules. Since Ref. 19 contains many different protocols for microtubule preparation, the authors should be more specific on how they made their microtubules. Please provide a more detailed protocol in the methods. Some control measurements with kinesin-1 motors would be helpful to confirm the mixture.

As suggested, we add a more detailed protocol in the methods (page 29 lines 524 – 529). We confirmed, slightly more exactly, the 5:3 ratio of our mixture of 13- and 14- protofilament with double-headed kinesin-1 in an in vitro microtubule gliding assay (Yajima et al. 2008). We revised the sentence to make it clearer (page 8 lines 114 – 124).

7. Page 7, line 13: The authors compare the handedness of the helical trajectories of MKLP1 with that of other kinesins and conclude that all of them turn leftward. However, as pointed out in Point 1 above, for kinesin-8, it is unclear whether sideward steps are random or have a bias.

We corrected the sentence. Now it reads (page 6 line 106 – page 7 line 111):

The handedness of the helical trajectories by MKLP1 was the same as that by other plus-end directed kinesins (when off-axis motion was detected) such as monomeric kinesin-1 (Yajima and

Cross 2005), kinesin-2 (Pan et al. 2010; Brunnbauer et al. 2012), kinesin-3 (Mitra et al. 2019), kinesin-5 (Yajima et al. 2008), and kinesin-8 (Bormuth et al. 2012; Mitra et al. 2018), and the opposite to that by a minus end-directed kinesin (kinesin-14 (Walker et al. 1990; Nitzsche et al. 2016)).

8. Are there single-molecule motility measurements with M2 and M2C2? Are speed and run length known for single molecules?

Our previous work (Hutterer et al. 2009) with surface-immobilized microtubules showed that speed and run length of dimeric ZEN-4 (585 aa)-GFP (corresponding to M₂) were 290 nm/s and 150 nm, respectively. Clustering drastically increases the run length (>800 nm) while slightly reducing the velocity (~210 nm/s). The velocity of the clustered M₂C₂ was 80 nm/s (Davies et al. 2015). Although we don't know its precise run-length, our rough impression from the kymographs (Davies et al. 2015) is that the association time seems to be comparable to that of M₂ and the run-length is likely to be shorter than that of M₂. Considering the physiological significance of the clustering, it would make sense only when we could determine these parameters as functions of the degree of clustering. This would need a lot of effort and is beyond the scope of our current work.

Reduction of the longitudinal velocity by CYK4-binding in our current assay is consistent with the above observations and those obtained in the surface gliding assay (White et al. 2013, Davies et al. 2015). We now mention the effect of CYK4-binding on the motility of the clustered MKLP1, which we failed to mention in the original manuscript, in Introduction (page 4, line 58) and when we report the reduction of the longitudinal velocity in our current assay (page 10, lines 152-154).

9. Fig. 2: Please change, 'Rotational revolutions' to 'Rotations' (also in Fig.

3). We have made corrections accordingly.

While the authors state the number of tested microspheres for the data shown, it is unclear, how many microspheres were tested in total. Did all microspheres that showed motility also show rotations? Did the authors discard traces? And if so, for what reasons?

Reliable measurement of the bead motion along and around the suspended microtubule requires a sufficiently long, continuous motion (we chose ? 1 μ m run length) along a single microtubule that is stably suspended without interference with other microtubules. Among 37 M₂, 63 M₂C₂ and 83 M₁ trajectories that satisfied these initial criteria, there were cases not suitable for further analyses, in which two beads came close to each other or the radius of projection to the y-z plane was too large (? 250 nm, i.e., bigger than twice as expected). After discarding these trajectories, we obtained 36, 47 and 47 trajectories for M₂, M₂C₂, and M₁, respectively. All of these were subjected for the analyses in this manuscript. Among them, 13

M₂C₂ and 6 M₁ were too short to complete one rotation around the microtubule. All the rest clearly showed net left-handed rotations. We now mention the above selection criteria for quality control in Methods (page 32, lines 595 – 599).

10. Fig. 3D: There is a typo in 'Rotational velocity'.

We have now corrected it.

11. Page 12: Power-law fitting of MSDs with non-integer exponents: Exponents $1 < \alpha < 2$ are attributed to anomalous diffusion which is not observed here. Power laws have a linear slope in a log-log plot, which is clearly not the case in Fig. 4B (apart fits and fit ranges are not shown or mentioned). Thus, there is no power law and anomalous diffusion here. It is a superposition of directed and random motion caused by the many motors attached to the microspheres. Such motion is well described by a parabola, which the authors used in Fig. S4. These fits should be shown in Fig. 4 on a log-log scale.

We agree that our discussion on the power-law was not to the point, and thus we removed the original Figure 4 from the revised manuscript.

Another reason for a non-linear slope of the log-log MSD plots, might be the presence of an offset expected from the tracking precision and data acquisition (see e.g. X. Michalet, Phys. Rev. E 82, 041914 (2010)). The authors should include an offset in their MSD fits. Please, also provide error bars on the fit parameters.

We really appreciate this reviewer's constructive criticisms and suggestions. By tracing the more recent papers that cited Michalet (2010), we have learned that there are technical subtleties in the traditional MSD analysis and various ways have been proposed to overcome them. Among novel developments, an approach to directly calculate the covariance-based estimator (CVE) of the diffusion coefficient (Vestergaard et al. 2014) has significant advantages over the traditional MSD fitting. 1) The CVE can be obtained by a simple calculation without curve fitting. Thus, we don't have to be bothered with the technical details of calculation of the MSD and optimization of the fitting. 2) The influences of the tracking precision and the motion blur are eliminated in principle through the calculation of the CVE that combines the variance of the displacements in the observation cycle and the covariance between the displacements in the neighboring time intervals. As expected from these features, it has been demonstrated that the CVE gives a robust estimate of the diffusion coefficient even for short trajectories for which the MSD fitting doesn't work very well (Vestergaard et al. 2015). We compared velocities and diffusion coefficients estimated by the MSD and CVE approaches in new Figure 4 H to M (plotted with error bars) and confirmed that the two approaches essentially agreed as to the difference caused by the motor constructs. Offsets in quadratic MSD fitting were zero with very small standard errors. This is mentioned in the legend of Figure 4.

An additional possibility for the nonlinear slopes, might be different populations of motors on the microspheres. For example, there might be a mixture of M2C2 with M2 that due to a low affinity of C2 are not present as a complex. Also, there might be different populations within M2C2. The authors could test whether the MSD values for a particular lag time are indeed normally distributed. Please show some histograms.

As discussed in this reviewer's point 5, according to the reported affinity of nM order between MKLP1(M2) and CYK4 (C2), we can expect that the majority of M2C2 remain as the complex without losing C2. We think it is reasonable to assume that the behavior of the M2C2-coated bead is primarily reflecting the characteristics of M2C2 instead of M2 (with minor contributions of C2-free M2). Indeed, the M2C2-coated beads showed the motility parameters clearly different from the M2-coated ones.

We thank this reviewer for suggesting to check the distribution of the MSD values to evaluate the homogeneities of our trajectory data. This prompted us to plot the 2-dimensional distribution of the longitudinal (X) and rotational (Y) displacements shown in new Figure 4 A to F. We have realized that, in addition to the roughly normally distributed main peak (arrow), a smaller population that stayed around the origin (arrowhead) was found for all the constructs. This turned out to be due to temporary pauses that randomly appeared in some trajectories (arrowheads in Figure 4G. For statics, please refer to Table S2). Although we currently don't know the reason why the MKLP1-coated beads sometime show a temporal pause, in most cases, the mean velocities didn't change before and after the pause (Figure 4G).

The above observation indicates that there is a heterogeneity in the modes of the bead motion driven by MKLP1, which would interfere with the calculation of the motility parameters defined under the assumption of the homogenous mode of motion, and forced us to re-examine our methods for analyzing the data. Although there is no direct method to know whether a bead was in the 'immobile' mode or 'mobile' mode for each time point, switching between the two modes seems to occur randomly with some probabilities. This allowed us to model the mode switching of the bead motion with a hidden Markov model with two states, 'mobile' mode in which the bead shows a positive mean displacement in a fixed lag time vs 'immobile' mode in which the mean displacement is zero. By a Bayesian inference with this model, we could objectively classify individual displacements into the two modes (Figure 4G, red: immobile, gray: mobile). Determination of the motility parameters (Figure H to M) and further analyses with our revised random walk model in Figure 5 were performed with the displacement data from the 'mobile' mode.

The random walk model has no justification and needs to be removed from the manuscript. It is essentially also a parabola and the fit parameters from Fig. S4 can simply be mapped onto the parameters of Eq. 1 & 2. There is no justification for the individual parameters of the model, e.g. the limitation of sideward steps to be diagonal, in particular since many motors interact

simultaneously. Also the exponent of 2 for the longitudinal motion is no justification. The longitudinal motion is hardly affected by pure sideward steps. A pure sideward step simply reduces the average forward speed, but does not affect the longitudinal exponent. It would be reasonable to exclude backward steps.

We fully agree with this reviewer that our mathematical model in the original manuscript was inappropriate, especially in limiting the sideward steps to be diagonal. Now, we have realized that trajectories of the beads on the 2-dimensional lattice surface contain the information about the rates or probabilities of the moves from the current site towards the eight neighbouring sites (four orthogonal plus four diagonal moves). These can be extracted by considering the temporal developments of the (higher-order) covariances between the displacements along the two dimensions, which can be estimated with an extended CVE approach (see equations in the ‘Inference of the hopping rates and preference’ section in Methods (pages 34 – 35)). Further mathematical details and evaluation by simulation of the robustness of the theory against various experimental factors such as the size of available data, the levels of noises, etc., have been described in a preprint deposited on arXiv (Mishima. arXiv:2008.12387).

Our fully revised analysis is based on this new approach and considers all the points raised by the reviewers including the geometry of the microtubule lattice surface (Reviewer #1) and the heterogeneity in the mode of motion (Reviewer #3). We could infer directional preferences of the bead motion, i.e., the probabilities to find the bead in each of the surrounding eight sites when it moved from the current site. This allowed us to compare the motion of the beads driven by the three MKLP1 constructs in a more intuitive way than comparing velocities and diffusion coefficients. Please refer to the revised main text pages 13 to 21 and Figures 4 and 5 for details.

Analysis with the random walk framework assumes that the bead moves on the 2-dimensional lattice making regular interaction on each step. The regular interaction doesn’t have to be via a pair of a single motor head and a single binding site on the microtubule surface but could be via multiple motors and multiple binding sites. Very strictly, however, this regularity is not guaranteed in our multiple-motor conditions, and thus, our analysis can’t be a perfect description of highly complicated, real phenomena. Rather it is an approximation with an imaginary bead that makes stochastic moves on the lattice with regular interaction. We believe this approximation is justified as the only way that is currently achievable to describe the features that can’t be addressed just by comparing the one-dimensional velocities and diffusion coefficients.

Based on the MSD fit parameters of Fig. S4, one can calculate the average stepping rates due to sideward diffusion, directed sideward motion, total number of sideward steps, forward steps per second, and the ratio of forward to total number of sideward steps as listed in the respective columns below for the different conditions (all values are in steps per second assuming 8 nm for a forward and 28 deg for a sideward step):

M2 13 4.6 18 35 2.0

M2C2 5.4 1.6 7 20 2.8

M1 12 1.3 13 28 2.1

Based on this table, the total number of sideward steps per second is less for the M2C2 complex compared to M2 (7 versus 18). Thus, the conclusion that CYK4 promotes the flexibility in off-axis stepping is not supported by the data. On the contrary, sideward stepping is reduced! The different behavior in the 2D gliding assays (Fig. 6), could also be explained by a difference in the processivity. Ref. 3 suggests that CYK4 changes the motor conformation to promote microtubule bundling preventing processive motion of the motor on a single microtubule. Thus, if M2 is more processive than M2C2, it would rotate microspheres to the left until the microsphere hits the surface and then follow along the microtubule axis without further apparent rotational diffusion. Since the rotational bias towards the left is larger for M2, rightward steps in an ensemble of motors may not be resolved. For M2C2, only single heads are able to interact with the microtubule limiting processivity.

Thus, there is more detachment/re-attachment kinetics that may cause the rotational diffusion. Therefore, the main conclusion of the manuscript is not supported by the data and the manuscript needs to be revised accordingly including the title. CYK4 binding seems to be important for the bundling of microtubules, but does not increase sideward stepping. Still obstacles may be avoided through detachment/re-attachment kinetics.

These arguments are based on a picture that the stochastic motion of a bead/motor can be fully described just with the four orthogonal moves and that the axial and sideward moves are independent processes, which we think is neither realistic nor proven. As we have discussed above, our new model considers the probabilities of the moves to all the eight neighboring sites, without assuming the independence between the two dimensions. This general theory covers the scenario that this reviewer has in mind as a special case. In the preprint deposited on arXiv (Mishima. arXiv:2008.12387), it was demonstrated by simulation that our approach can distinguish between two random walks that show exactly the same velocities and diffusion coefficients but have different directional preferences (with or without diagonal steps) under the experimental conditions similar to our current case (data sizes, precisions etc.).

12. Fig. 5D, right: The '60' on the time axis is misaligned.

We have removed the old Figure 5.

13. Fig. 6F: For consistency, also add the M2 and M2C2 nomenclature to the subplot. As suggested by Reviewer #1 as well, we have removed the old Figure 6F.

14. Page 20, Table 1: Please provide raw data and standard errors for ATPase activity

parameters K_{cat} and K_M .

As requested, we have now included raw data of our microtubule-stimulated ATPase activity as a plot in Supplementary Figure S5 and added the standard errors for ATPase activity parameters in Table 1.

Please provide the total number of tested microspheres and list how many moved less than 1 micron, how many moved more than 1 micron and did/did not rotate. In particular the latter number is important. Please show exemplary traces of microspheres that did not rotate and include them in the data analysis of Fig. 2-4.

Please see our response to the comment 9. All the trajectories we obtained showed rotational motions although some trajectories were too short to complete one full cycle. All of them were included into our analyses.

15. The pitch determination is unclear. Did the authors divide the total distance by the total number of rotations of an individual trace?

We now added the description about the pitch determination (page 7 Figure 2B legend and page 32 line 599 – page 33 line 611). Please also refer to our response to Reviewer #1's other point 1). A sentence about the pitch determination of the corkscrewing motions of microtubules in the surface gliding assay shown in Figure S3 was also added (page 37, lines 702 – 703).

16. The discussion needs to be revised.

We revised the entire Discussion section (pages 24 – 27). After a quick overview of our findings, we discuss the possible mechanisms for torque generation or biased protofilament switching by monomeric motors and by dimeric motors in comparison with various mechanisms proposed for the axial motion. We then discuss the possible mechanisms for the modulation of the MKLP1's torque generation by CYK4, mentioning the caution about interpretation of our results due to the involvement of multiple motors. After brief comments on the heterogeneity in the mode of motion by MKLP1 and on our novel modeling approach, we discuss the possible biological roles of the off-axis motion and torque generation by centralspindlin. Please also refer to our response to Reviewer #1's comments.

17. Page 31: Please provide more details on the quantum dots.

We add information about the quantum dots we used in Methods ("Qdot 525 Streptavidin Conjugate, Thermo Fisher Scientific", page 37 lines 694 – 695).

18. Page 32: Please state how microspheres were washed. The different motor concentrations for incubation suggest that different surface densities may have been present during experiments. Was the observed motility independent of the incubation concentrations?

In the ‘‘Preparation of kinesins-coated bead solutions’’ section in Methods, we added information about the volumes of the bead–motor mixture (4 μl) and the assay buffer (100 μl) used for resuspension of the sedimented beads (page 31, line 570 and 571) without extra washing. This provides the basis for the calculation in the above point 5. We used the minimum motor:bead ratio that support the stable motility along the microtubule. If prepared with lower motor:bead ratio, beads didn’t show stable motility. We don’t have data for the higher motor:bead ratios. However, we observed a similar trend in the pitches of the corkscrewing motion of the microtubules in the surface gliding assay (Figure S3) that the pitches by M₂C₂ and M1 is larger than that by M₂.

19. Fig. S1: Please, also provide the precision for the radial position. How does the experimentally measured radius compare to the 140 nm estimate? Also, please, provide a reference or short comment about the radial and angular variance equations.

We estimated the precisions of the angular positions using the well-known equation of the propagation of uncertainty. For $f = f(u, v)$, the variance of f is

$$\sigma^2(f) = \left(\frac{\partial f}{\partial u}\right)^2 \cdot \sigma^2(u) + \left(\frac{\partial f}{\partial v}\right)^2 \cdot \sigma^2(v)$$

when the random variables, u and v , are independent of each other.

The angular position θ is determined by (y, z) as $\theta = \tan^{-1}(z/y)$. As a first-order approximation, it is reasonable to assume that the random errors in the measurement of y and z are not correlated to each other. Thus, using

$$\begin{aligned} \frac{\partial \theta}{\partial y} &= -(\cos \theta)^2 \cdot \frac{z}{y^2} \\ \frac{\partial \theta}{\partial z} &= (\cos \theta)^2 \cdot \frac{1}{y} \end{aligned}$$

the variance of θ can be estimated by

$$\begin{aligned} \sigma^2(\theta) &= \left(\frac{\partial \theta}{\partial y}\right)^2 \sigma^2(y) + \left(\frac{\partial \theta}{\partial z}\right)^2 \sigma^2(z) \\ &= (\cos \theta)^4 \cdot \frac{z^2}{y^4} \cdot \sigma^2(y) + (\cos \theta)^4 \cdot \frac{1}{y^2} \cdot \sigma^2(z) \\ &= (\cos \theta)^2 \cdot \frac{(\cos \theta)^2}{y^2} \cdot (\tan \theta)^2 \cdot \sigma^2(y) + (\cos \theta)^2 \cdot \frac{(\cos \theta)^2}{y^2} \cdot \sigma^2(z) \\ &= (\cos \theta)^2 \cdot \frac{(\cos \theta)^2}{y^2} \cdot (\tan \theta)^2 \cdot \sigma^2(y) + (\cos \theta)^2 \cdot \frac{(\cos \theta)^2}{y^2} \cdot \sigma^2(z) \\ &= \frac{\sigma^2(y) \sin^2 \theta + \sigma^2(z) \cos^2 \theta}{r^2} \end{aligned}$$

This indicates that the precision of the angular position can be evaluated with the precisions of y and z , and the value of r , and that the precision of r is not necessary as long as the central axis

is reasonably correctly specified (please also refer to our response to the below point 20). Considering the geometry of the spherical bead around the microtubule axis, it is reasonable to assume that r is largely constant. We estimated $r = \sim 140$ nm, based on the radius of microtubule, sizes of the MKLP1 constructs and antibodies, and the radius of the microtubule. This value is consistent with the actually observed radius of the helical trajectories (Figures 2A and S2). With this value and the measured $\sigma(y)$ and $\sigma(z)$, the estimate of $\sigma(\theta)$ varies between 5° and 7.5° depending on the angle (Figure S1D). Since r can't be smaller than the sum of the radii of the microtubule and the bead, $r \geq \sim 125$ nm. Thus, using $r = 125$ nm, the worst case estimate of the precision of θ would be $5.6^\circ \sim 8.4^\circ$, which remains smaller than the angle between two protofilaments.

To avoid unnecessary distraction, we removed the formula and plot about $\sigma(r)$ from Figure S1. In the legend, we clarified that the formula is based on the general law of error propagation.

20. Fig. S2B: The tracks appear to be rather elliptical suggesting that the 3D tracking calibration was not very accurate.

As this reviewer pointed out, the projection of the trajectory of M₂ in Figure S2B appear to be slightly elliptical. This is because the suspended microtubule was not perfectly parallel to the x -axis but slightly tilted due to the limitation in the precision of manufacturing the glass stages. A close look into the z -coordinates (blue line) in Figure S2A finds that the peaks and troughs gradually drift down ~ 0.05 μm in z per 5 μm along x .

The angle of the tilt is very small ($\tan^{-1}(0.05 \mu\text{m} / 5 \mu\text{m}) = 0.6^\circ$ in this case) and thus the shape of the y - z section of the microtubule (and the ideally helical trajectory around it) can still be treated as a circle. The influence of the tilted microtubule appears as the z -drift of the circle in the y - z section, which becomes more evident towards the ends of a long trajectory. The measured angular position, ϕ , is

$$\phi = \tan^{-1} \left[\frac{\alpha + \sin \theta}{\cos \theta} \right]$$

where θ is the true angular position and α is the relative z -drift = (z -drift) : (radius of the helical trajectory). As shown in the plots below, even in the extreme case of $\alpha = 0.3$ or -0.3 , which corresponds to ~ 140 nm $\times 0.3 \times 2 = 84$ nm difference in the z -coordinate between the ends of a trajectory, i.e., $\tan^{-1}(84 \text{ nm} / 5 \mu\text{m}) = 1^\circ$ tilt for a $5 \mu\text{m}$ trajectory, the deviation of ϕ from θ is smaller than the angle for a single proto-filament (yellow and blue curves. Dotted gray lines

represent the periodicity of the protofilaments). In most of our cases, the tilt of the microtubule was much smaller. Thus, we did not perform any special correction for this minor irregularity.

Reviewers' comments:

Reviewer #1 (Remarks to the Author):

The manuscript has been improved from the prior submission through significant changes to the introduction and discussion. The authors addressed my major concern about motor ensembles by clarifying, at many places throughout the text, that beads contain an unknown number of motor complexes.

I also had a concern about the physiological relevance of centralspindlin motility in general and in the rebuttal the authors noted: "Accumulation of wild-type centralspindlin to the tips of equatorial astral microtubules has also been reported (Nishimura & Yonemura 2006).

In addition to Nishimura and Yonemura, which reported imaging of fixed samples, plus-end localization and tracking of Kinesin-6 has been observed in living cells by Vale, Spudich and Griffis (JCB, 2009) as well as Verma and Maresca (eLife, 2019). It is important to note that kinesin-6 (Pavarotti) motility on astral MTs was not evident in either study. While this may be a difference between *Drosophila* and human Kinesin-6, it does call into question the authors comment that: "These clearly indicate the physiological importance of the efficient transport of centralspindlin." Even in Hutterer, Glotzer and Mishima (CB 2009) where motility is reported in human cells, the motility events appeared rare and quite short run lengths. I raise this issue to re-emphasize my original point that stepping around obstacles by ensembles may not be critically important for a non-processive complex that makes mostly short runs to the plus-end.

Despite my concerns about the physiological significance of the authors' "obstacle" hypothesis, I feel the manuscript has been improved and I support its publication.

Reviewer #2 (Remarks to the Author):

I am satisfied with the answers to my suggestions and I have no further remarks. I think the revisions in response to all the comments have significantly improved the manuscript. In my opinion, the revised manuscript is suitable for publication.

Reviewer #3 (Remarks to the Author):

The revised manuscript by Maruyama and coauthors has improved a lot. I appreciate the effort that the authors have dedicated to the revision. Many of my concerns have been addressed to a satisfactory degree. However, I still maintain my concern about the simulation and still think it would be better to omit it. Instead, the authors should focus on their experimental results and e.g. include/show the (higher-order) covariance data. The authors argue correctly that this data provides evidence for coupling between the degrees of freedom (forward vs. sideward motion). It would be better to demonstrate this coupling instead of simulating a scenario that does neither correspond to their experimental assay nor to the *in vivo* situation that involves clusters of motors. This latter point needs to be stressed even more in the introduction/motivation and discussion. Even though it is mentioned now, it does not become that apparent in the manuscript that it is essential for the biological function. The aspect of multiple motors on the microspheres is now pointed out more frequently. However, there is no discussion on any multimotor effects or alternative interpretations. There is no discussion on the detachment / reattachment and potential reorientation of the microsphere, e.g. as a means to bypass obstacles. The short single motor run lengths should be explicitly mentioned since they imply that single motors indeed detached and reattached many times during a microsphere run. Also, motors exert forces onto each other. There is no need to apply an

external force on the microsphere. There will always be one motor that is lagging behind the others. This motor will experience assisting forces while the others experience hindering loads. The same argument holds true for sideward forces. Thus, the discussion is still rather focussed on single-motor behavior instead of a collective of motors. An additional concern that I have with respect to the simulation is that the authors assume now eight different binding sites corresponding to the neighboring tubulin dimers. However, it is not even known how large the step size for a single motor dimer is. With a single neck linker length of about 70 amino acids corresponding to a contour length of about 25 nm, the dimer has a reach of up to 50 nm! This long reach implies that individual motor heads may be separated by several tubulin dimers. The bottom line is that there are many possibilities for potential binding sites. Thus, the eight binding sites is an assumption in the simulation that not even for a single dimer may be justified. Furthermore, the authors use the same parameters also for the monomeric motors that presumably are not processive, do not take "8-nm steps", and obviously rely on a multimotor scenario. Thus, as I pointed out above, it would be more convincing to provide evidence for the coupling between the degrees of freedom by showing the covariance data and providing an interpretation of the covariance data in context of a multimotor assay. In conclusion, while the manuscript has much improved, I think it still should be revised once more to address my above concerns.

Some minor points:

1. L202: correct sentence/grammar
2. L210: No data is provided that shows that not only forward but also sideward motion stopped.
3. If authors insist on keeping the simulation, some parts should go in the methods/supplement, assumptions and limitations of the interpretations should be clearly stated. I strongly do not recommend this approach.
4. The affinity data should be correlated with the experimental data and not the simulation.
5. P23: It is unclear what the errors are.
6. P24: I would remove claims of novelty.

Our response to the reviewers' comments

Original Reviewers' comments

Our response

Reviewer #1 (Remarks to the Author):

The manuscript has been improved from the prior submission through significant changes to the introduction and discussion. The authors addressed my major concern about motor ensembles by clarifying, at many places throughout the text, that beads contain an unknown number of motor complexes.

I also had a concern about the physiological relevance of centralspindlin motility in general and in the rebuttal the authors noted: "Accumulation of wild-type centralspindlin to the tips of equatorial astral microtubules has also been reported (Nishimura & Yonemura 2006).

In addition to Nishimura and Yonemura, which reported imaging of fixed samples, plus-end localization and tracking of Kinesin-6 has been observed in living cells by Vale, Spudich and Griffis (JCB, 2009) as well as Verma and Maresca (eLife, 2019). It is important to note that kinesin-6 (Pavarotti) motility on astral MTs was not evident in either study. While this may be a difference between Drosophila and human Kinesin-6, it does call into question the authors comment that: "These clearly indicate the physiological importance of the efficient transport of centralspindlin." Even in Hutterer, Glotzer and Mishima (CB 2009) where motility is reported in human cells, the motility events appeared rare and quite short run lengths. I raise this issue to re-emphasize my original point that stepping around obstacles by ensembles may not be critically important for a non-processive complex that makes mostly short runs to the plus-end.

We appreciate that this reviewer kindly notified us that we failed to cite Verma and Maresca (2019). It is now cited in the current manuscript.

Despite my concerns about the physiological significance of the authors' "obstacle" hypothesis, I feel the manuscript has been improved and I support its publication.

We thank this reviewer for supporting the publication of our manuscript.

Reviewer #2 (Remarks to the Author):

I am satisfied with the answers to my suggestions and I have no further remarks. I think the revisions in response to all the comments have significantly improved the manuscript. In my opinion, the revised manuscript is suitable for publication.

We thank this reviewer for supporting the publication of our manuscript.

Reviewer #3 (Remarks to the Author):

The revised manuscript by Maruyama and coauthors has improved a lot. I appreciate the effort that the authors have dedicated to the revision. Many of my concerns have been addressed to a satisfactory degree. However, I still maintain my concern about the simulation and still think it would be better to omit it. Instead, the authors should focus on their experimental results and e.g. include/show the (higher-order) covariance data. The authors argue correctly that this data provides evidence for coupling between the degrees of freedom (forward vs. sideward motion). It would be better to demonstrate this coupling instead of simulating a scenario that does neither correspond to their experimental assay nor to the in vivo situation that involves clusters of motors.

We are pleased to hear that we properly addressed many of this Reviewer's concerns although, regrettably, failed to fully clarify the assumptions and limitations in our mathematical modeling. We wish to maintain Figure 5 as it is since, as was the case for the traditional diffusion coefficients in Figure 4, the individual higher-order parameters (now presented in Supplementary Figure 5) per se are not very informative in discussing the differences in the motion of the beads driven by different constructs. The hopping rates and probabilities, which are easier to understand, are linearly linked to the higher-order parameters under the assumption that the bead motion is a 2D random walk and can be directly determined by the trajectory data. On the other hand, as this reviewer correctly pointed out, we can't strictly exclude the possibilities of the modes of bead motion that don't fit with the realm of random walk although we believe that the random hopping to the neighboring sites is a reasonable assumption for mathematical simplicity. Thus, in this version, we rewrote the first paragraph of the "Modeling the stochastic helical motion as a biased random walk on the MT surface lattice" section (page 11) to clarify our motivation and the assumptions and limitations of our model and modified the discussion.

"To get better insights on the fluctuating helical motions of the beads driven by the MKLP1 constructs, we attempted to describe them as a 2D random walk, assuming that the bead moves by stochastic hopping to the nearby binding sites on the microtubule lattice surface at the anisotropic/biased rates. This is a mathematical simplification since we can't strictly exclude the possibilities against our assumption such as jumping beyond the neighboring sites. However, as shown below, comparison of the beads motion by different constructs on this scheme lead us to interesting results." (pages 11-12, lines 252-257)

"Although we assumed that the bead moves on the microtubule lattice by hopping to the neighboring sites, we need to keep in mind that this a mathematical simplification and might not strictly be the case. For example, the bead might jump beyond the neighboring sites by rapid detachment and attachment. Nonetheless, by comparing the beads motion by different constructs

with our new method, we could obtain important insights on the hopping preferences,” (page 17, lines 392-395)

This latter point needs to be stressed even more in the introduction/motivation and discussion. Even though it is mentioned now, it does not become that apparent in the manuscript that it is essential for the biological function.

We modified the introduction (pages 3-4, lines 55-57), which now reads “Higher-order clustering promotes the plus-end directed transport without detachment from the microtubule and is essential for timely equatorial accumulation, which is crucial for cytokinesis signaling”

The aspect of multiple motors on the microspheres is now pointed out more frequently. However, there is no discussion on any multimotor effects or alternative interpretations. There is no discussion on the detachment / reattachment and potential reorientation of the microsphere, e.g. as a means to bypass obstacles.

The detachment and reattachment is a possible way of bypassing the obstacles. However, this might compromise the transport.

The short single motor run lengths should be explicitly mentioned since they imply that single motors indeed detached and reattached many times during a microsphere run.

We now mention the short run-lengths of the single MKLP1 dimers following our observations of the requirement of the multiple motors for the stable motion of the beads, “This is consistent with the short run lengths of the MKLP1 constructs when they are not assembled into the higher-order clusters (Hutterer 2009 and Davies 2015).” (page 5, lines 94-95).

Also, motors exert forces onto each other. There is no need to apply an external force on the microsphere. There will always be one motor that is lagging behind the others. This motor will experience assisting forces while the others experience hindering loads. The same argument holds true for sideward forces. Thus, the discussion is still rather focussed on single-motor behavior instead of a collective of motors.

We agree that, upon discussing the possible mechanisms, we could only refer to the papers based on single-motor behaviors. However, this was mainly because we failed to find a published piece of work that explains biased motions by the multiple-motor effect. We agree with this reviewer that, under the multiple-motor condition, we can't exclude the possibility of mechanical communication between the motors (although, for non-processive motors such as MKLP1, this effect wouldn't be as strong as the case for processive motors such as kinesin-1). However, considering that the arrangement of the motors on the bead surface is totally random, we are afraid that, the explanation of the torque generation by the multiple motor effect would

be too speculative. We clarified this point in the current version, “Although we can’t exclude the possible mechanical communication between multiple motors on a bead, this effect alone would not be able to explain the torque generation since the arrangement of the motors on a bead or coverglass surface should be random.” (page 19, lines 440-444 in Discussion).

An additional concern that I have with respect to the simulation is that the authors assume now eight different binding sites corresponding to the neighboring tubulin dimers. However, it is not even known how large the step size for a single motor dimer is. With a single neck linker length of about 70 amino acids corresponding to a contour length of about 25 nm, the dimer has a reach of up to 50 nm! This long reach implies that individual motor heads may be separated by several tubulin dimers. The bottom line is that there are many possibilities for potential binding sites. Thus, the eight binding sites is an assumption in the simulation that not even for a single dimer may be justified.

Although MKLP1 has a long neck region, at least a part of it is folded into a globular domain and the distance between the two heads in the MKLP1 dimer measured in solution (weakly associated with the mica surface) was 13 ± 4 nm (Davies et al. 2015), much shorter than the reviewer’s estimation assuming complete unfolding. Although we can’t strictly exclude the possibility of simultaneous binding of the two heads on the distant binding sites (16 nm or longer apart) beyond the neighboring sites, we think that such an event must be relatively rare.

Furthermore, the authors use the same parameters also for the monomeric motors that presumably are not processive, do not take "8-nm steps", and obviously rely on a multimotor scenario. Thus, as I pointed out above, it would be more convincing to provide evidence for the coupling between the degrees of freedom by showing the covariance data and providing an interpretation of the covariance data in context of a multimotor assay. In conclusion, while the manuscript has much improved, I think it still should be revised once more to address my above concerns.

As described above, we believe that modeling with a 2D random walk with biased hopping to the neighbouring sites should be acceptable as the first approximation.

Some minor points:

1. L202: correct sentence/grammar

“... were located slightly deviated off...” was corrected to be “... were slightly deviated off...”

2. L210: No data is provided that shows that not only forward but also sideward motion stopped.

Now it reads “(Figure 4g for the longitudinal motion, sections marked with arrowheads on the original trajectories. Supplementary Figure 4 for the rotational motion)”(page 9, lines 201-203).

3. If authors insist on keeping the simulation, some parts should go in the methods/supplement, assumptions and limitations of the interpretations should be clearly stated. I strongly do not recommend this approach.

As mentioned above, we wish to keep the modeling in Figure 5. The assumptions and limitations are stated in both Results and Discussions.

4. The affinity data should be correlated with the experimental data and not the simulation.

We believe that the comparison is reasonable. The hopping probabilities were determined by inference from the experimental data on the reasonable assumption of 2D random walk.

5. P23: It is unclear what the errors are.

The values are 'mean \pm SD'. This is now clarified.

6. P24: I would remove claims of novelty.

We removed "for the first time" as pointed out. (page 17, line 380)

REVIEWERS' COMMENTS:

Reviewer #3 (Remarks to the Author):

The authors now have sufficiently motivated and framed the simulation and added some useful hints pointing towards alternative interpretations of the data. Thus, I support its publication.

PS.: In the introduction, it might be helpful to point out the distance between heads of the MKLP1 dimer and that the neck linkers form a globular domain.

Our response to the reviewers' comments

Original Reviewers' comments

Our response

Reviewer #3 (Remarks to the Author):

The authors now have sufficiently motivated and framed the simulation and added some useful hints pointing towards alternative interpretations of the data. Thus, I support its publication.

We thank this reviewer for supporting the publication of our manuscript.

PS.: In the introduction, it might be helpful to point out the distance between heads of the MKLP1 dimer and that the neck linkers form a globular domain.

We now mention that the neck region might be folded into a compact structure and the inter-head distance is about 13 nm (page 3, lines 39-40 and lines 53-54).